# Motor neurons are dispensable for the assembly of a sensorimotor circuit for gaze stabilization

Dena Goldblatt[1,2], Basak Rosti[1], Kyla Rose Hamling[1], Paige Leary[1], Harsh Panchal[1], Marlyn Li[1,2], Hannah Gelnaw[1], Stephanie Huang[1,2], Cheryl Quainoo[1], David Schoppik[1]*

[1]Department of Otolaryngology, Neuroscience & Physiology, and the Neuroscience Institute, NYU Grossman School of Medicine, New York, United States; [2]Center for Neural Science, New York University, New York, United States

## eLife Assessment

This **important** study asks whether motor neurons within the vestibulo-ocular circuit of zebrafish are required to determine the identity, connectivity, and function of upstream premotor neurons. They provide **compelling** and comprehensive genetic, anatomical and behavioral evidence that the answer is, "No!". This work will be of general interest to developmental neurobiologists and will motivate future studies of whether motor neurons are dispensable for assembly of other sensorimotor neural circuits.

*For correspondence:
schoppik@gmail.com

**Abstract** Sensorimotor reflex circuits engage distinct neuronal subtypes, defined by precise connectivity, to transform sensation into compensatory behavior. Whether and how motor neuron populations specify the subtype fate and/or sensory connectivity of their pre-motor partners remains controversial. Here, we discovered that motor neurons are dispensable for proper connectivity in the vestibular reflex circuit that stabilizes gaze. We first measured activity following vestibular sensation in pre-motor projection neurons after constitutive loss of their extraocular motor neuron partners. We observed normal responses and topography indicative of unchanged functional connectivity between sensory neurons and projection neurons. Next, we show that projection neurons remain anatomically and molecularly poised to connect appropriately with their downstream partners. Lastly, we show that the transcriptional signatures that typify projection neurons develop independently of motor partners. Our findings comprehensively overturn a long-standing model: that connectivity in the circuit for gaze stabilization is retrogradely determined by motor partner-derived signals. By defining the contribution of motor neurons to specification of an archetypal sensorimotor circuit, our work speaks to comparable processes in the spinal cord and advances our understanding of principles of neural development.

## Introduction

Developing sensorimotor reflex circuits must precisely connect functional subtypes of neurons to ensure appropriate behavior. For example, withdrawal from noxious stimuli requires maturation of a sensorimotor circuit that uses subtypes of spinal interneurons to transform noxious stimulation into activation of both ipsilateral flexor and contralateral extensor motor neurons (*Sherrington, 1906*). Work over the past 40 years has highlighted motor partner populations as possible orchestrators of connectivity in pre-motor reflex circuits (*Matise and Lance-Jones, 1996*; *Glover, 2003*; *Arber, 2012*;

**eLife digest** Some external stimuli, such as a painful touch or sudden head movements, can trigger automatic physical responses. These reactions are controlled by sensorimotor circuits which are comprised of three types of neurons. First, sensory neurons detect the external stimulus. They then pass the information to interneurons, which relay the signal to motor neurons that activate the muscles required to produce a prompt physical response.

Sensorimotor circuits form very early in life, but it remains unclear how the three types of neurons involved contribute to one another's development. Previous research suggests that motor neurons send chemical signals 'upstream' to interneurons to help them mature. However, there is conflicting evidence both for and against this hypothesis.

To investigate this theory, Goldblatt et al. studied a sensorimotor circuit found in all vertebrates known as the gaze stabilization reflex. The sensory neurons in this circuit detect head movements and transmit this information, via interneurons, to the motor neurons that control muscles in the eye. This allows gaze to remain stable while the head moves.

Goldblatt et al. used a genetic tool to eliminate the motor neurons involved in the gaze stabilization reflex from zebrafish larvae. This prevented the motor neurons from sending chemical signals upstream to the interneurons and caused the zebrafish larvae to develop eyes that were permanently rotated outward.

However, further experiments revealed that the connections between sensory neurons and interneurons still developed normally despite the absence of motor neurons. The interneurons also expressed the appropriate set of genes for the stage of larval development tested. This suggests that interneurons involved in the gaze stabilization reflex can develop normally without chemical signals from motor neurons.

These findings are a major step towards understanding how sensorimotor circuits develop, and suggest that current models of this process may need to be revised. It remains to be seen whether other sensorimotor circuits can also develop without signals from downstream motor neurons. Gaining more detailed insights into how these circuits develop could also enhance our understanding of certain neurological conditions further down the line.

---

*Dasen, 2017*; *Ladle et al., 2007*; *Wan et al., 2019*), but controversy remains about the nature of their role. In the spinal cord, molecular perturbations of motor neuron identity have provided evidence both for (*Machado et al., 2015*, *Hinckley et al., 2015*, *Vrieseling and Arber, 2006*, *Baek et al., 2017*, *Balaskas et al., 2019*) and against (*Wang and Scott, 2000*, *Sürmeli et al., 2011*, *Bikoff et al., 2016*, *Sweeney et al., 2018*, *Shin et al., 2020*) an instructive role in establishing connectivity. Part of this controversy stems from the wide variety of inputs to spinal motor neurons (*Kiehn, 2016*), the molecular and functional heterogeneity of pre-motor interneurons (*Bikoff et al., 2016*, *Sweeney et al., 2018*), and their complex roles in gait and posture (*Hunt, 1923*). Further, transcription factors play multivariate and redundant roles in spinal motor neuron development (*Sharma et al., 1998*, *Hutchinson and Eisen, 2006*), such that the effects of molecular perturbations of identity can be masked.

The sensorimotor circuit for vertical gaze stabilization offers a simple framework to evaluate whether and how motor neurons shape pre-motor circuit fate and connectivity. The vertebrate vestibulo-ocular reflex circuit consists of three neuron types – peripheral sensory, central projection, and extraocular motor neurons – that stabilize gaze after head/body tilts (*Figure 1A*; *Szentágothai, 1964*). Subtype fate, anatomical connectivity, and function are inextricably linked: directionally-tuned sensory neurons innervate nose-up/nose-down subtypes of projection neurons, which in turn innervate specific motor neurons that selectively control either eyes-down or eyes-up muscles (*Straka et al., 2001*; *Diaz et al., 2003*; *Glover, 2003*; *Bianco et al., 2012*; *Schoppik et al., 2017*; *Bagnall and Schoppik, 2018*; *Diaz and Puelles, 2019*; *Liu et al., 2022*; *Goldblatt et al., 2023*). As both the recipients and origin of directional information, projection neuron fate specification is paramount to proper circuit assembly. Recent work has established the vertical vestibulo-ocular reflex circuit in zebrafish as a model to uncover determinants of fate and connectivity (*Bianco et al., 2012*, *Schoppik et al., 2017*, *Bagnall and Schoppik, 2018*, *Goldblatt et al., 2023*) given the ease

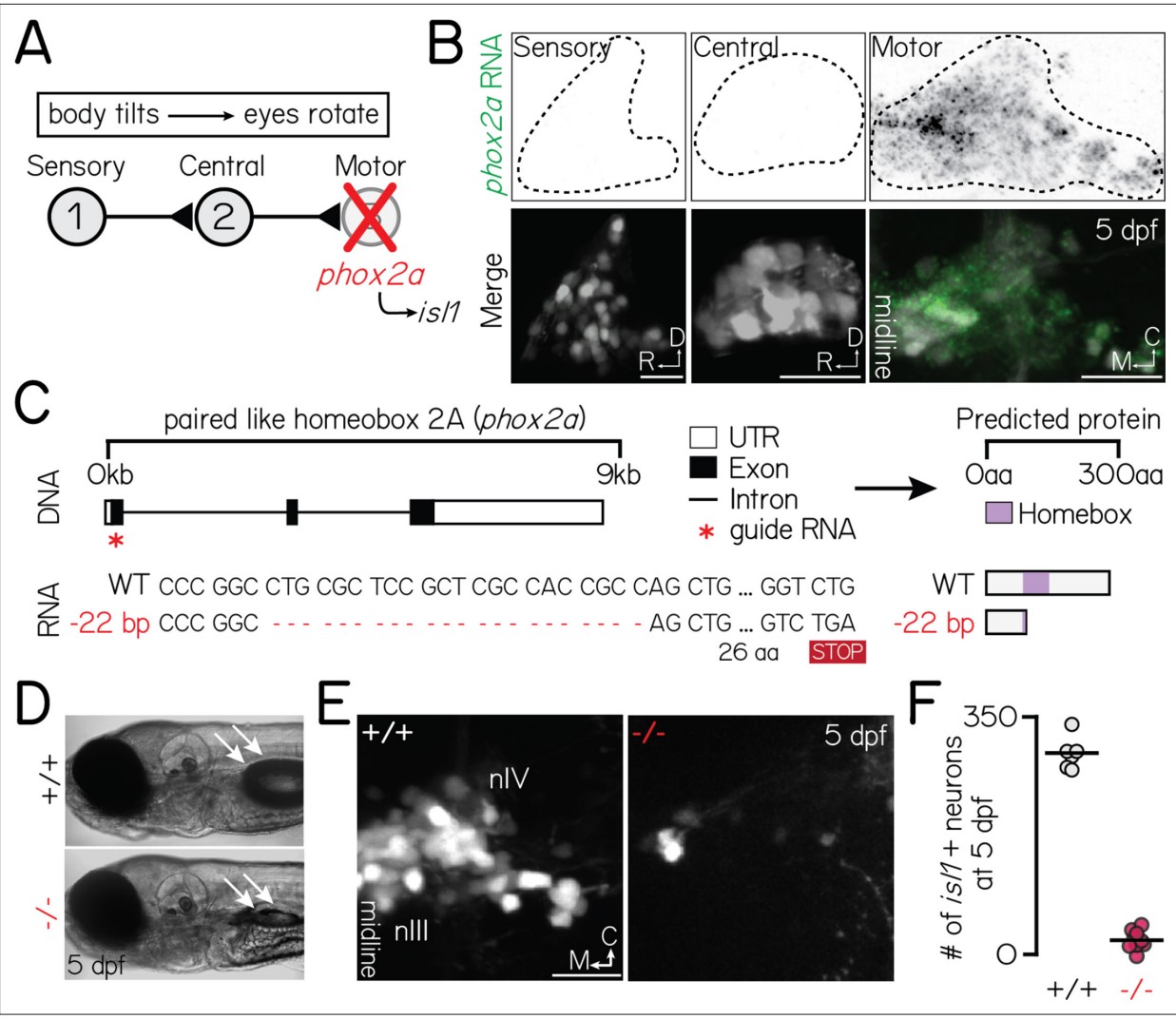

**Figure 1.** *phox2a* loss-of-function mutants fail to develop nIII/nIV motor neurons and vertical eye rotation behavior. Associated with *Figure 1—figure supplement 2*. (**A**) Schematic of vestibulo-ocular reflex circuitry and the genetic loss-of-function approach used to perturb motor-derived signals. (**B**) Fluorescent in situ hybridization showing *phox2a* transcript expression in statoacoustic ganglion sensory afferents (left), central projection neurons in the tangential nucleus (middle), or nIII/nIV extraocular motor neurons (right) at 5 days post-fertilization (dpf). Top: probe only, nuclei outlined with dashed lines. Bottom: probe (green) merged with somata, labeled by *Tg(–6.7Tru.Hcrtr2:GAL4-VP16);Tg(UAS-E1b:Kaede)* (sensory, central) or *Tg(isl1:GFP)* (motor). (**C**) Schematic of CRISPR/Cas9 mutagenesis approach. Top: Red star shows location of guides against *phox2a* DNA. Bottom: RNA sequence in wildtype and *phox2a^d22* alleles. Red dashed lines show deleted sequence; 'STOP' box shows predicted premature stop codon due to deletion. Right shows predicted protein sequence. (**D**) Transmitted light image of a 5 dpf wildtype (top) and *phox2a* null mutant (bottom). White arrows point to a normally inflated (top) or absent (bottom) swim bladder. (**E**) Images of nIII/nIV motor neurons in one hemisphere, labeled by *Tg(isl1:GFP)*, in wildtype siblings (left) and *phox2a* null mutants (right) at 5 dpf. Scale bar, 20 μm. (**F**) Quantification of the number of *Tg(isl1:GFP)*+neurons in nIII/nIV from N=6 wildtype siblings and N=10 *phox2a* null mutants.

The online version of this article includes the following figure supplement(s) for figure 1:

**Figure supplement 1.** Extraocular motor neuron fate is perturbed in *phox2a* mutants.

**Figure supplement 2.** phox2a specifies nIII motor neuron fate in a dose- and birthdate-dependent manner.

of optical imaging, abundant tools for genetic perturbations, rapid development, and robust evolutionary conservation.

The current model for vestibulo-ocular reflex circuit development was motivated by pioneering work in chick (*Glover, 2000*, *Glover, 2003*) and formalized by Hans Straka: "[circuit assembly] is accomplished by a specification process that retrogradely transmits post-synaptic target identities to pre-synaptic neurons." (*Straka, 2010*). In its strongest form, this 'retrograde' model posits a causal role

for extraocular motor neurons in specifying the fate (sensory selectivity) of central projection neurons. This key prediction – that loss of motor neurons would disrupt sensory selectivity in pre-synaptic projection neurons – remains untested. In zebrafish, extraocular motor neurons are temporally poised for such a role. Motor neurons are organized into spatial pools, and though synaptogenesis at ocular muscle targets begins late in development (*Clark et al., 2013*), motor neuron fate (muscle target and pool location) is determined early (*Greaney et al., 2017*). Projection neurons are born at roughly the same time as motor neurons and extend axons shortly afterwards, poising them to receive deterministic signals that could retrogradely specify their sensory selectivity (*Goldblatt et al., 2023*).

Here, we adopted a loss-of-function approach to determine whether motor partner populations specify identity or instruct connectivity across an entire vestibular reflex circuit in zebrafish. We generated a new mutant allele for the *phox2a* gene to eliminate the extraocular motor neurons used for vertical gaze stabilization. Combining functional, anatomical, and sequencing approaches, we then demonstrated that motor neurons are dispensable for three aspects of pre-motor reflex circuit assembly: (i) directionally-appropriate connectivity between sensory and projection neurons, (ii) assembly of projection neurons with motor partners, and (iii) the transcriptional profiles of projection neurons. The current model of vestibulo-ocular reflex circuit development must therefore be revised: up/down projection neuron subtype fate cannot be retrogradely established by a motor partner-derived signal. Instead, the signals that specify fate must lie elsewhere. More broadly, our work argues against a deterministic role of motor neurons in specifying the fate and sensory connectivity of pre-motor circuit components.

## Results

### Constitutive loss of *phox2a* prevents extraocular motor neuron specification and impairs vertical gaze stabilization behavior

Extraocular motor neurons for vertical/torsional gaze stabilization are located in cranial nuclei III (nIII) and IV (nIV). To eliminate nIII/nIV motor neurons and by extension, any secreted signals, we used a genetic loss-of-function approach (*Figure 1A*). A single highly-conserved transcription factor, *phoxa*, specifies nIII/nIV fate (*Guo et al., 1999*; *Coppola et al., 2005*; *Hasan et al., 2010*; *Nakano et al., 2001*). In the vestibulo-ocular reflex circuit, *phox2a* is exclusively expressed in nIII/nIV motor neurons but not its upstream partners (*Figure 1B*). Therefore, *phox2a* is an ideal genetic target to eliminate motor-derived signals without compromising evaluations of upstream functional development.

Prior mutagenesis established a *phox2a* loss-of-function allele in zebrafish (*Guo et al., 1999*), but the line has since been lost. Here, we generated three new *phox2a* loss-of-function alleles using CRISPR/Cas9 mutagenesis (*Figure 1C*; one allele shown here; additional alleles described in Materials and methods). Consistent with prior reports and human mutations (*Bosley et al., 2006*), both eyes in *phox2a* null mutants were exotropic (rotated towards the ears) reflecting a loss of motor neurons in nIII/nIV. *phox2a* mutants failed to hatch from their chorions without manual intervention and did not inflate their swim bladders by 5 days post-fertilization (dpf; *Figure 1D*), phenotypes not previously reported (*Guo et al., 1999*). Consequently, null mutants do not survive past 7 dpf. We did not observe these morphological phenotypes in wildtype and heterozygous siblings (*Figure 1D*). As vestibulo-ocular reflex circuit architecture and behavior is established by 5 dpf (*Bianco et al., 2012*; *Schoppik et al., 2017*; *Goldblatt et al., 2023*), premature lethality did not preclude further measurements of circuit development.

To validate *phox2a* loss-of-function, we leveraged a downstream transcription factor: *isl1* (*Varela-Echavarría et al., 1996*). The *Tg(isl1:GFP)* line (*Higashijima et al., 2000*) labels all nIII/nIV motor neurons except inferior oblique neurons (*Greaney et al., 2017*), which comprise one of four pools for upwards eye rotations. We first quantified changes in the number of labeled nIII/nIV neurons (*Figure 1E, F*). In *phox2a* mutants, we observed an expected and near-total loss of *isl1* expression (WT: 298±19 neurons across both hemispheres; null: 19±11 neurons; Wilcoxon rank sum test, p=2.5 × $10^{-4}$) at 5 dpf, well-after nIII/nIV differentiation is complete (*Greaney et al., 2017*). As a secondary measure of nIII/nIV motor neuron fate, we performed fluorescent in situ hybridization against *vachta*, a marker of cholinergic motor neurons, when differentiation is complete at 2 dpf. Unlike sibling controls, nIII/nIV neurons in *phox2a* mutants failed to express *vachta*, validating that loss of *phox2a* perturbs at least two key transcriptional markers of nIII/nIV fate (*Figure 1—figure supplement 1*). Unexpectedly,

we also observed slightly fewer neurons in *phox2a* heterozygotes (heterozygote: 229±20 neurons; Wilcoxon rank sum test against WT, p=6.7 × 10⁻⁴). In heterozygotes, loss of *isl1* fluorescence was restricted to the medial domain of dorsal nIII, which contains some of the earliest-born neurons in nIII/nIV (*Figure 1—figure supplement 2A–C*; *Greaney et al., 2017*). Globally, this manifested as a rostral and ventral shift in the positions of all neurons mapped (*Figure 1—figure supplement 2D*; two-sample, two-tailed KS test, WT vs. heterozygotes: mediolateral axis, p=0.13; rostrocaudal: p=4.0 × 10⁻²⁹; dorsoventral: p=2.5 × 10⁻⁹). This region contains two motor pools that control the inferior (IR) and medial rectus (MR) muscles (*Greaney et al., 2017*). We conclude that *phox2a* acts in a dose- and birthdate-dependent manner to specify nIII motor pool fate.

Together, these observations validate our *phox2a* loss of function alleles as a selective means to disrupt nIII/nIV motor neuron fate specification and vertical eye rotation behavior.

## Peripheral-to-central circuit assembly does not require motor partners

Vertical gaze stabilization requires that (1) peripheral VIII[th] nerve sensory afferents relay tilt sensation (nose-up/nose-down) directly to projection neurons in the tangential nucleus, and (2) projection neurons innervate appropriate nIII/nIV pools (eyes-up/eyes-down). For proper circuit function, appropriate connectivity must first develop across up/down circuit subtypes. The 'retrograde' model predicts that motor partners specify circuit assembly. Therefore, in the absence of motor neurons, projection neurons should fail to respond selectively to directional tilt sensations – either due to loss of their fate, the fate of upstream sensory afferents, or sensory-to-central connectivity.

To evaluate upstream circuit formation, we measured tilt-evoked responses in projection neurons using Tilt-In-Place Microscopy (TIPM) (*Hamling et al., 2023*; *Figure 2A, B*). Peripherally, tilts activate utricular VIII[th] nerve sensory inputs to projection neurons (*Hamling et al., 2023*; *Goldblatt et al., 2023*). We used a galvanometer to deliver tonic nose-up and nose-down pitch tilts to *phox2a* null larvae and sibling controls. We then measured the activity of a calcium indicator, GCaMP6s (*Chen et al., 2013*), in projection neurons. We performed experiments at 5 dpf, when nearly all projection neurons are selective for one tilt direction (*Goldblatt et al., 2023*), circuit architecture is stable (*Schoppik et al., 2017*), and gaze stabilization behavior is directionally-appropriate (*Bianco et al., 2012*).

Projection neuron responses and topography were strikingly unchanged in *phox2a* mutants compared to controls. There was no change in the number of projection neurons observed across N=5 sibling controls and *phox2a* mutants (mean control number: 42 ± 4 neurons per hemisphere; mean in *phox2a* mutants: 39 ± 3 neurons; two-tailed Wilcoxon rank sum test, p=0.22). We next recorded the activity of n=297 neurons from N=16 *phox2a* mutants and n=440 neurons from N=21 sibling controls (Materials and methods and *Table 1* split by genotype). We observed comparable ratios of projection neuron subtypes (sib: 46% nose-down, 47% nose-up, 7% untuned; *phox2a*: 49% nose-down, 44% nose-up, 7% untuned; *Figure 2C*). Next, we evaluated their topography (*Figure 2D, E*). Projection neurons are topographically organized along the dorso-ventral axis by their directional selectivity (*Goldblatt et al., 2023*). Global spatial separation between subtypes remained significant in *phox2a* mutants (one-way multivariate ANOVA, p=0.004). We also compared the topography of nose-up and nose-down neurons separately across *phox2a* genotypes (Materials and methods and *Table 2* split by genotype). Nose-down neurons were comparably distributed between null and control larvae (one-way multivariate ANOVA, p=0.15). We observed a minor lateral shift to nose-up neurons in null mutants (median mediolateral position, sib: 15.2 µm from medial edge; *phox2a*, 13.2 µm; two-tailed, two-sample KS test, p=3.0 × 10⁻⁴) but no changes in other spatial axes (dorsoventral: p=0.16; rostrocaudal: p=0.56). The small medial deviation (2 µm across a 40 µm space) is within the limits of our registration error. We conclude that projection neuron topography is established independently of motor partners.

Projection neuron sensitivity and selectivity also developed comparably between *phox2a* mutants and siblings (*Figure 2F–K*). Projection neurons responded to tilt sensations with comparable magnitudes (*Figure 2G–J*) (nose-down mean ΔF/F, sib: 1.86±1.69; *phox2a*: 2.07±1.48; two-tailed Wilcoxon rank sum test, p=0.98; nose-up mean ΔF/F, sib: 1.24±1.23; *phox2a*: 1.02±0.89; p=0.18). Previously, we defined a metric to describe a neuron's selectivity for one tilt direction (0=equal responses to up/down; 1=maximally selective) *Goldblatt et al., 2023*. Directional selectivity remained unchanged in *phox2a* mutants (*Figure 2H–K*) (nose-down mean index, sib: 0.73±0.29; *phox2a*: 0.68±0.29; two-tailed

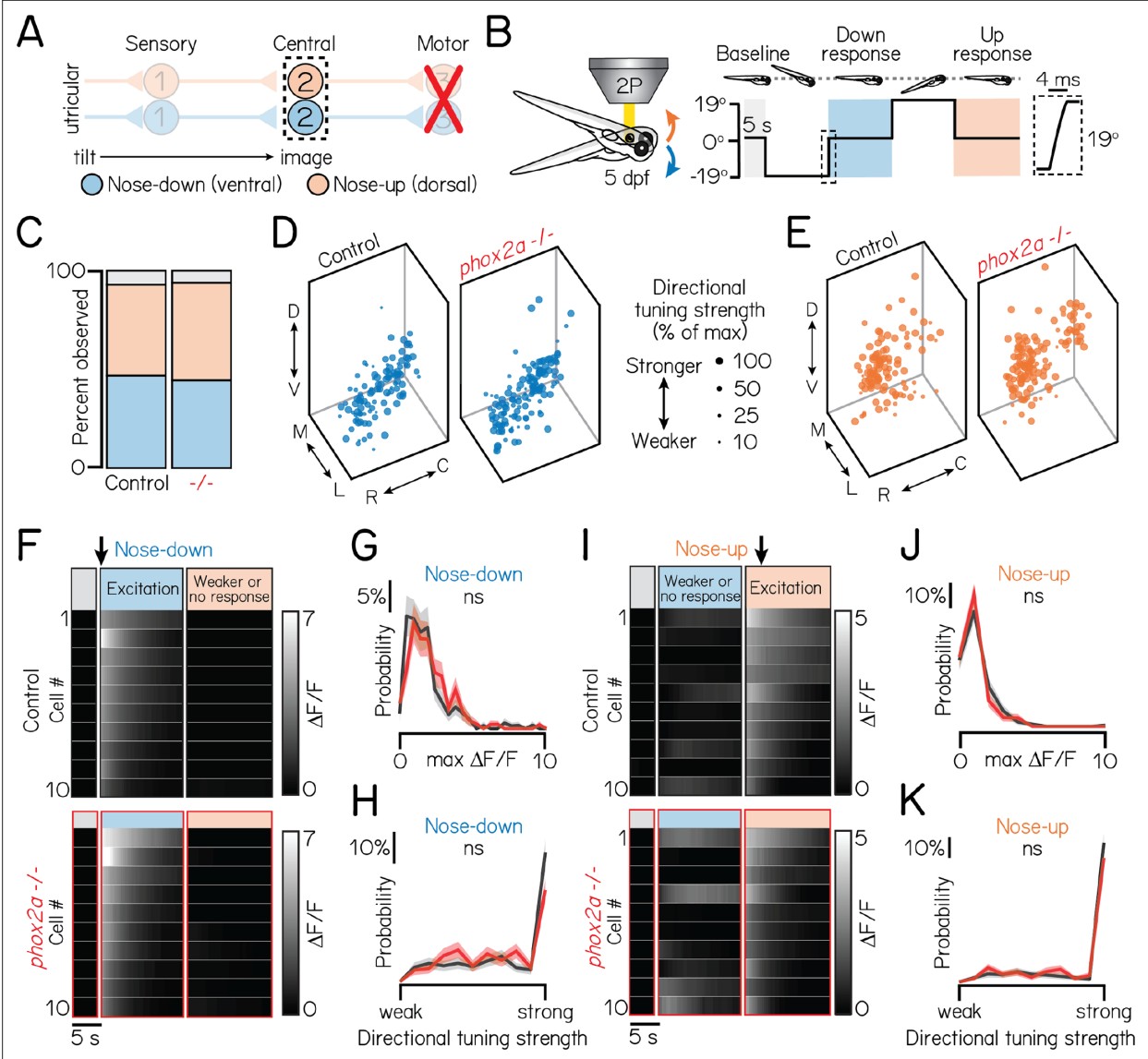

**Figure 2.** Motor neurons are dispensable for proper connectivity between utricular sensory afferents and projection neurons. Associated with *Table 1*. (**A**) Schematic of pitch vestibulo-ocular reflex circuitry. Dashed lines outline projection neurons as calcium imaging target. Nose-down/eyes-up channel represented with blue; orange, nose-up/eyes-down. (**B**) Schematic of tonic pitch-tilt stimulus delivered with Tilt-In-Place Microscopy (TIPM). Shaded regions show calcium imaging windows when fish were oriented horizontally immediately following tilts. Inset shows timecourse of the rapid step to restore horizontal position after tilts. Imaging experiments used larvae from the *Tg(isl1:GFP);Tg(–6.7Tru.Hcrtr2:GAL4-VP16);Tg(UAS:GCaMP6s)* line. (**C**) Proportion of subtypes observed in sibling controls and *phox2a* null mutants. Blue: nose-down. Orange: nose-up. Grey: Neurons without directional tuning (criteria in Materials and methods). (**D, E**) Soma position of nose-down (blue) and nose-up (orange) neurons in sibling controls (left) and *phox2a* null mutants (right). Soma size scaled by strength of directional selectivity (min = 0; max = 1; see Materials and methods). (**F/I**) Heatmaps showing example tilt responses from nose-down (**F**) or nose-up (**I**) neurons in sibling controls (top) and *phox2a* null mutants (bottom). n=10 neurons with strongest ΔF/F responses to tilts shown. Each row shows an individual neuron. Shaded bars show calcium imaging window immediately following restoration from eccentric position. Black arrow points to first second of tilt response used for analyses. (**G/J**) Distributions of maximum ΔF/F responses to tilts for nose-down (**G**) or nose-up (**J**) neurons in sibling controls (black) and *phox2a* null mutants (red). Solid and shaded lines show mean and standard deviation, respectively, of bootstrapped data (Materials and methods). (**H/K**) Distributions of directional tuning score to tilts for nose-down (**H**) or nose-up (**K**) neurons in sibling controls (black) and *phox2a* null mutants (red). Tuning score ranges from 0 (equal responses to both tilt directions, no tuning) to 1 (responses to one tilt direction only); criteria detailed in Materials and methods. Solid and shaded lines show mean and standard deviation, respectively, of bootstrapped data.

**Table 1.** Statistical comparisons of tilt responses across genotypes.

WT (sampled) refers to an n=125 neuron subset, sampled with replacement from a reference dataset of wildtype projection neurons. Data shown is mean/standard deviation unless otherwise noted. p val generated from a one-way ANOVA with multiple comparisons. Associated with *Figure 2* and *Figure 3*.

| | WT (all) | WT (sampled) | phox2a+/+ | phox2a+/- | phox2a-/- | p val |
|---|---|---|---|---|---|---|
| Tonic tilt stimuli | | | | | | |
| n (neurons/fish) | 255/10 | 125 /x | 76/5 | 109/6 | 297/16 | |
| % observed (nose-up/nose-down/untuned) | 50/44/7 | 37/54/9 | 40/54/7 | 56/37/7 | 44/50/6 | |
| ΔF/F, nose-up | 1.28 ± 1.23 | 1.27 ± 1.19 | 1.09 ± 1.03 | 1.12 ± 0.90 | 1.02 ± 0.82 | 0.26 |
| ΔF/F, nose-down | 2.01±1.66 | 1.99 ± 1.69 | 1.38 ± 0.91 | 1.98 ± 1.61 | 2.07±1.48 | 0.16 |
| directional tuning strength, nose-up | 0.84 ± 0.28 | 0.83 ± 0.30 | 0.87 ± 0.26 | 0.81 ± 0.28 | 0.81 ± 0.29 | 0.70 |
| directional tuning strength, nose-down | 0.72 ± 0.30 | 0.72 ± 0.31 | 0.73 ± 0.31 | 0.77 ± 0.30 | 0.68 ± 0.29 | 0.54 |
| Impulse stimuli | | | | | | |
| n (neurons/fish) | 255/10 | 125 /x | 76/5 | 109/6 | 297/16 | |
| % observed (responsive/unresponsive) | 58/42 | 57/43 | 57/43 | 60/39 | 70/30 | |
| ΔF/F | 0.41 ± 0.46 | 0.33 ± 0.28 | 0.29 ± 0.29 | 0.22 ± 0.16 | 0.32 ± 0.28 | 1.0E-05 |
| directional tuning strength | 0.08 ± 0.36 | 0.10 ± 0.38 | 0.003 ± 0.41 | 0.07 ± 0.48 | 0.07 ± 0.41 | 0.64 |
| Multiple comparisons | genotype | | p val | Cohen's d | | |
| ΔF/F to impulses | WT to sampled | | p=0.13 | 0.21 | | |
| | WT to +/+ | | p=0.04 | 0.27 | | |
| | WT to +/- | | p=3.8E-06 | 0.48 | | |
| | WT to -/- | | p=0.006 | 0.24 | | |
| | +/+to +/- | | p=0.47 | 0.34 | | |
| | +/+to -/- | | p=0.89 | 0.11 | | |
| | +/-to -/- | | p=0.02 | 0.49 | | |

Wilcoxon rank sum test, p=0.85; nose-up mean index, sib: 0.85±0.26; *phox2a*: 0.81±0.29; p=0.12). Collectively, this demonstrates that the functional responses of projection neurons and, by inference, connectivity with utricular afferents are not shaped by motor partners.

Ventral projection neurons receive additional input from the semicircular canals (*Goldblatt et al., 2023*), which encode phasic (fast) tilt sensation. To activate sensory afferents from the semicircular canals, we used TIPM to deliver two impulses of angular rotation (*Figure 3A, B* ; *Hamling et al., 2023*; *Goldblatt et al., 2023*). We observed no changes in *phox2a* mutants (*Table 1* and *Table 2* for genotype splits). Projection neurons responded to impulses in comparable ratios (*Figure 3E*) (sib: 58% responsive; *phox2a*: 71% responsive). Responsive projection neurons remained localized to the ventral nucleus (dorsoventral axis: two-tailed, two-sample KS test, p=0.99). We observed minor positional changes in the rostrocaudal and mediolateral axes (*Table 2*), which are not topographically organized by impulse responsivity *Goldblatt et al., 2023*; this deviation is again within our margin of registration error. Lastly, the functional properties of projection neurons were unchanged (*Figure 3F, G*). We observed no change in calcium response magnitudes (*Figure 3F*) (mean ΔF/F, sib: 0.33±0.29; *phox2a*: 0.36±0.40; two-tailed Wilcoxon rank sum test, p=0.85) or lack of directional selectivity (*Figure 3G*) (mean index, sib: 0.07±0.41; *phox2a*: 0.06±0.40; p=0.39). Therefore, fate and connectivity between phasic sensory afferents and projection neurons must not require motor partners.

Lastly, we considered whether loss of one subtype of nIII/nIV neurons might alter connectivity. For example, loss of eyes-down motor pools could impair wiring between their corresponding nose-up sensory and projection neuron partners. Here, we leveraged *phox2a* heterozygotes, which lack a subtype of nIII neurons (IR/MR) that contribute to downwards eye rotations (*Figure 1—figure*

**Table 2.** Statistical comparisons of projection neuron topography across genotypes.

WT (sampled) refers to an n=125 neuron subset, sampled with replacement from a reference dataset of wildtype projection neurons. Data shown is the median/standard deviation distance from the ventro-lateral and rostral edges of the tangential nucleus (total size: 40 µm across each spatial axis). p val from one-way ANOVA (individual spatial axes) or multivariate ANOVA (global organization), respectively.

| | WT (all) | WT (sampled) | phox2a+/+ | phox2a+/- | phox2a-/- | p val |
|---|---|---|---|---|---|---|
| Tonic tilt stimuli | | | | | | |
| *nose-down* | | | | | | |
| n (cells/fish) | 111/10 | 51 /x | 41/5 | 40/6 | 147/16 | |
| dorsoventral | 20.0 ± 8.4 | 25.0 ± 8.4 | 15.0 ± 9.0 | 25.0 ± 8.7 | 20.0 ± 9.1 | 0.03 |
| mediolateral | 17.4 ± 6.0 | 17.6 ± 6.5 | 19.3 ± 6.6 | 15.3 ± 5.6 | 16.8 ± 6.6 | 0.15 |
| rostrocaudal | 16.8 ± 9.6 | 16.5 ± 9.6 | 19.5 ± 9.6 | 16.1 ± 10.4 | 18.2 ± 9.6 | 0.49 |
| global organization | | | | | | 0.09 |
| *nose-up* | | | | | | |
| n (cells/fish) | 111/10 | 51 /x | 41/5 | 40/6 | 147/16 | |
| dorsoventral | 30.0 ± 10.3 | 25.0 ± 10.8 | 27.5 ± 10.9 | 25.0 ± 10.4 | 30.0 ± 10.5 | 0.67 |
| mediolateral | 14.9 ± 7.1 | 15.1 ± 5.9 | 16.7 ± 7.0 | 18.6 ± 9.8 | 13.2 ± 7.2 | 9.6E-06 |
| rostrocaudal | 16.1 ± 9.8 | 16.7 ± 10.4 | 12.5 ± 10.7 | 14.8 ± 10.4 | 14.8 ± 10.7 | 0.51 |
| global organization | | | | | | 2.7E-07 |
| Multiple comparisons | genotype | | p val | Cohen's d | | |
| dorsoventral, nose-down | WT to sampled | | 0.86 | 0.17 | | |
| | WT to +/+ | | 0.54 | 0.29 | | |
| | WT to +/- | | 0.96 | 0.13 | | |
| | WT to -/- | | 0.08 | 0.34 | | |
| | +/+to +/- | | 0.35 | 0.41 | | |
| | +/+to -/- | | 0.99 | 0.05 | | |
| | +/-to -/- | | 0.07 | 0.45 | | |
| mediolateral, nose-up | WT to sampled | | 0.91 | 0.03 | | |
| | WT to +/+ | | 0.57 | 0.35 | | |
| | WT to +/- | | 0.006 | 0.57 | | |
| | WT to -/- | | 0.69 | 0.20 | | |
| | +/+to +/- | | 0.74 | 0.23 | | |
| | +/+to -/- | | 0.09 | 0.55 | | |
| | +/-to -/- | | 5.4E-06 | 0.74 | | |
| Impulse stimuli | | | | | | |
| n (responsive) | 148/10 | 138 /n | 43/5 | 55/5 | 214/16 | |
| dorsoventral | 20.0 ± 9.8 | 20.0 ± 10.1 | 20.0 ± 10.7 | 20.0 ± 10.9 | 20.0 ± 10.3 | 0.57 |
| mediolateral | 19.0 ± 5.9 | 16.1 ± 6.8 | 16.7 ± 8.6 | 20.4 ± 7.2 | 19.8 ± 5.8 | 0.003 |
| rostrocaudal | 21.9 ± 9.9 | 17.3 ± 9.8 | 15.4 ± 9.1 | 13.0 ± 9.9 | 22.0 ± 9.5 | 2.9E-05 |
| global organization | | | | | | 1.2E-0.9 |
| Multiple comparisons | genotype | | p val | Cohen's d | | |
| mediolateral, responsive | WT to sampled | | 0.99 | 0.002 | | |

*Table 2 continued on next page*

*Table 2 continued*

| | WT (all) | WT (sampled) | phox2a+/+ | phox2a+/- | phox2a-/- | p val |
|---|---|---|---|---|---|---|
| | WT to +/+ | | 0.32 | 0.37 | | |
| | WT to +/- | | 9.82 | 0.17 | | |
| | WT to -/- | | 0.33 | 0.23 | | |
| | +/+to +/- | | 0.91 | 0.14 | | |
| | +/+to -/- | | 0.007 | 0.55 | | |
| | +/-to -/- | | 0.05 | 0.37 | | |
| rostrocaudal, responsive | WT to sampled | | 0.98 | 0.37 | | |
| | WT to +/+ | | 0.12 | 0.29 | | |
| | WT to +/- | | 0.001 | 0.53 | | |
| | WT to -/- | | 0.07 | 0.10 | | |
| | +/+to +/- | | 0.90 | 0.72 | | |
| | +/+to -/- | | 0.95 | 0.33 | | |
| | +/-to -/- | | 0.23 | 0.21 | | |

*supplement 2*). We observed no differences in tonic tilt responses between *phox2a* wildtype, heterozygote, and null larvae, though we did note a minor decrease in response strength to impulses (statistics in *Table 1*). We note that *phox2a* heterozygotes do not lack all motor pools for downwards eye rotations. Nevertheless, we conclude that individual motor pools do not meaningfully contribute to connectivity between sensory and projection neurons.

Taken together, these experiments demonstrate intact directional selectivity for two peripheral sensory inputs – utricular and semicircular canal VIII[th] nerve afferents – and appropriate connectivity with projection neurons. We conclude that functional sensory-to-central circuit formation is established independently of motor partners.

## Projection neurons remain competent to assemble with appropriate motor targets

Motor partners could secrete signals that initiate pre-motor axon outgrowth, target arriving axons to specific motor pools, or trigger synaptogenesis (*Glover, 2003*). Motor pool topography in nIII/nIV reflects ocular muscle targets: dorsal pools innervate downward-rotating muscles (superior oblique and inferior rectus), while ventral pools target the converse (eyes-up, superior rectus and inferior oblique; *Evinger, 1988*, *Greaney et al., 2017*). In turn, projection neuron somatic and axonal organization mirrors motor pool topography (*Liu et al., 2022*, *Goldblatt et al., 2023*), which could facilitate directionally-selective circuit assembly. We reasoned that projection neurons may fail to initiate axon outgrowth, target spatially-appropriate motor pools, and/or form synapses in *phox2a* mutants. To test this hypothesis, we measured changes in projection neuron anatomy at 5 dpf, when axonal arbors are established and stable (*Schoppik et al., 2017*).

To test whether projection neurons establish gross, long-range (hindbrain to midbrain) axonal outgrowths, we performed optical retrograde labeling (*Pujala and Koyama, 2019*) using a photolabile protein, Kaede. We targeted the medial longitudinal fasciculus at the midbrain-hindbrain boundary, which contains projection neuron axons (*Schoppik et al., 2017*, *Goldblatt et al., 2023*; *Figure 4A*). In both *phox2a* mutants and sibling controls, we observed retrograde photolabeling of projection neuron soma (*Figure 4B*), supporting that initial axon outgrowth does not require motor partner-derived signals.

Next, we evaluated whether projection neuron axons remain capable of wiring with spatially-appropriate motor partners. Projection neuron axons segregate along the dorsal (nose-up) and ventral (nose-down) axes according to their birth order (early/late born, respectively; *Liu et al., 2022*, *Goldblatt et al., 2023*) and the pool topography of their motor targets (*Greaney et al., 2017*).

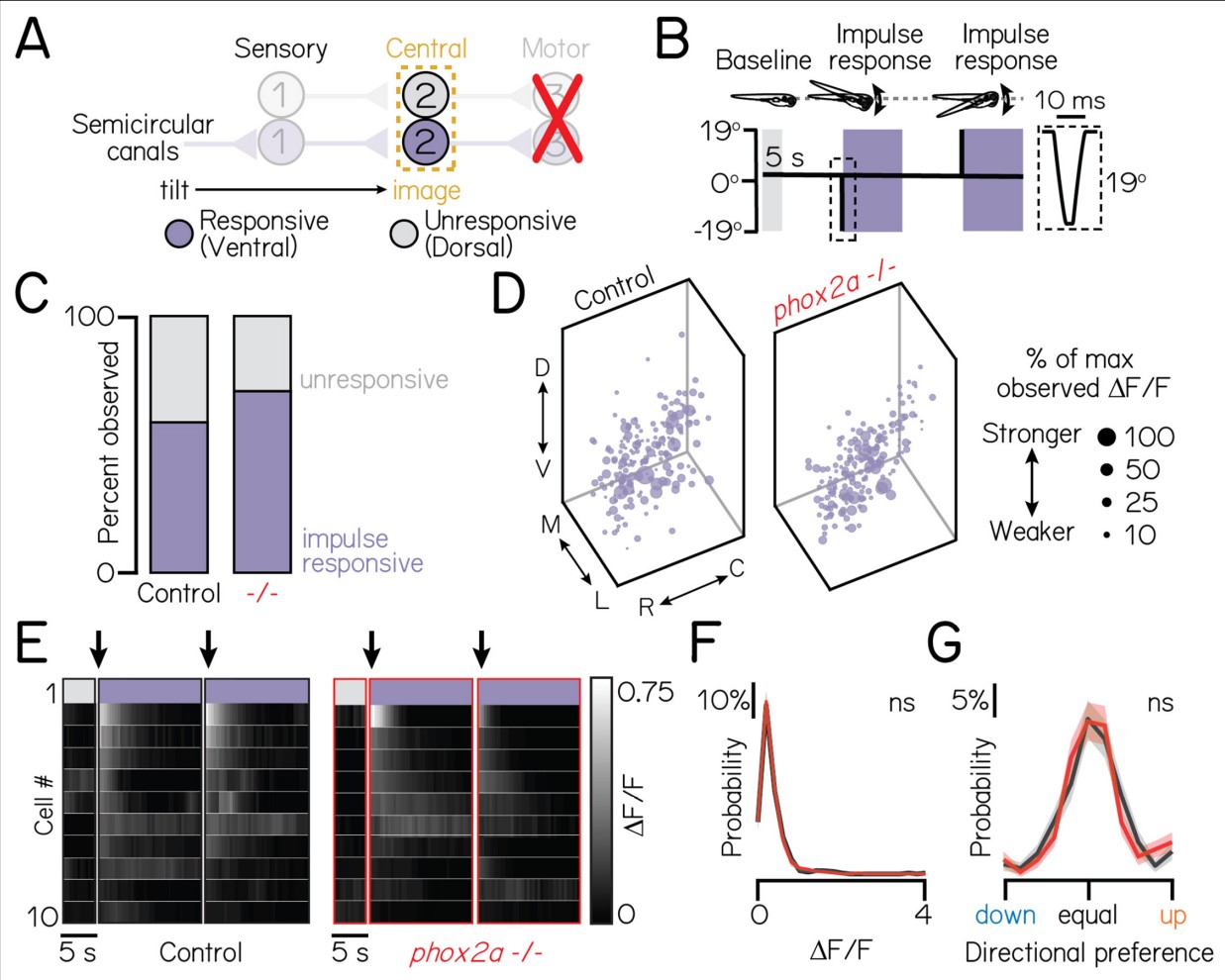

**Figure 3.** Motor neurons are dispensable for proper connectivity between semicircular canal sensory afferents and projection neurons. Associated with **Table 1**. (**A**) Schematic of impulse tilt experiments. Yellow dashed lines outline projection neurons as calcium imaging target. Impulse-responsive neurons (ventrally-localized) shown with purple; unresponsive neurons, grey. (**B**) Schematic of impulse stimuli delivered with TIPM. Shaded regions show calcium imaging windows at horizontal immediately following impulses. Inset shows timecourse of impulse stimulus. Imaging experiments used larvae from the *Tg(isl1:GFP);Tg(–6.7Tru.Hcrtr2:GAL4-VP16);Tg(UAS:GCaMP6s)* line. (**C**) Proportion of impulse-responsive (purple) and unresponsive (grey) neurons observed in sibling controls and *phox2a* null mutants. (**D**) Soma position of impulse-responsive neurons in sibling controls (left) and *phox2a* null mutants (right). Soma size scaled by strength of calcium response (ΔF/F), normalized by max observed ΔF/F. (**E**) Heatmaps showing example impulse responses from neurons in sibling controls (left) and *phox2a* null mutants (right). n=10 example neurons shown. Each row shows an individual neuron. Shaded bars show calcium imaging window immediately following impulse delivery. Black arrow points to first second of tilt response used for analyses. Note smaller scale (0–0.75) of impulse responses relative to **Figure 2F and I**. (**F**) Distributions of maximum ΔF/F responses to impulses in sibling controls (black) and *phox2a* null mutants (red). Solid and shaded lines show mean and standard deviation, respectively, from bootstrapped data. (**G**) Distributions of directional tuning score to impulses in sibling controls (black) and *phox2a* null mutants (red). Tuning score ranges from 0 (equal responses to both tilt directions, no tuning) to 1 (responses to one tilt direction only); criteria detailed in Materials and methods. Solid and shaded lines show mean and standard deviation, respectively, from bootstrapped data.

To test whether projection neurons retain this topography, we optically labeled the axons of early-born (before 30 hpf) projection neurons (***Goldblatt et al., 2023***). In *phox2a* mutants, axons remained dorsoventrally segregated at midbrain targets (***Figure 4C***, inset). Typically, projection neurons robustly collateralize to nIII/nIV targets at the midbrain-hindbrain boundary. We did not observe collaterals to nIII/nIV in *phox2a* mutants (***Figure 4C***). However, projection neurons still robustly arborized to more rostral, spinal-projecting targets in the nucleus of the medial longitudinal fasciculus, suggesting they retain the machinery necessary to collateralize. Consistent with this hypothesis, we observed that projection neurons formed occasional, small collaterals in *phox2a* mutants with few (1–5%) nIII/nIV

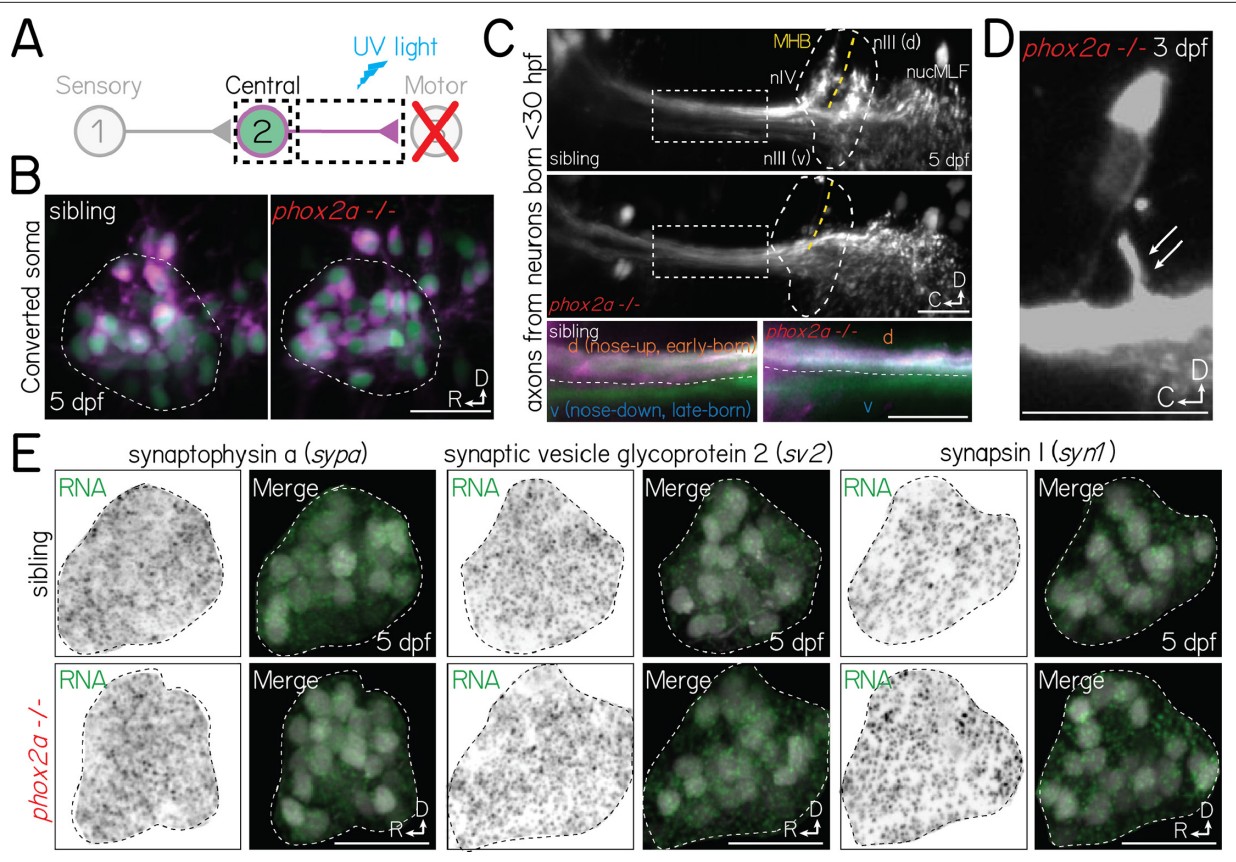

**Figure 4.** Projection neurons are anatomically and molecularly poised to assemble with motor neuron partners in *phox2a* mutants. (**A**) Schematic of retrograde photofill experiments. Projection neuron axons expressing the photolabile protein Kaede are targeted at the midbrain-hindbrain boundary with ultraviolet light. Converted protein (magenta) retrogradely diffuses to the soma, while the unconverted nucleus remains green. (**B**) Projection neuron somata in sibling controls (left) and *phox2a* null mutants (right) after retrograde photolabeling. Experiments performed at 5 dpf. Neurons visualized in *Tg(isl1:GFP);Tg(–6.7Tru.Hcrtr2:GAL4-VP16);Tg(UAS:E1b-Kaede)*. (**C**) Top two panels: Projection neuron axons at the hindbrain (inset) and midbrain-hindbrain boundary in sibling controls (top) and *phox2a* null mutants (bottom). Axons visualized using *Tg(isl1:GFP);Tg(–6.7Tru.Hcrtr2:GAL4-VP16);Tg(UAS:E1b-Kaede)*. White dashed outline (circle) shows arborization fields in nIII/nIV. Dashed box over axons shows location of two bottom panels. MHB and yellow dashed line, midbrain-hindbrain boundary. nucMLF: nucleus of the longitudinal fasciculus. Bottom two panels: Zoom of axons (dashed rectangle above). Spatial segregation between early-born (magenta +green) and late-born (green only) axons. White dashed line reflects separation between dorsal (nose-up, early-born) and ventral (nose-down, late-born) axon bundles. Image at 5 dpf in sagittal view. (**D**) Projection neuron axon bundle in a *phox2a* null mutant at 3 dpf. White arrows point to single collateral to two nIII/nIV neurons that were not eliminated following *phox2a* knockout. (**E**) Fluorescent in situ hybridization against RNA for three pre-synaptic markers: synaptophysin a (*sypa*; left), synaptic vesicle glycoprotein 2 (*sv2*, middle), and synapsin I (*syn1*, right). Top row, sibling controls. Bottom row, *phox2a* null mutants. For each panel set, left images show in situ probe expression (green) and right images show merge with projection neurons labeled in *Tg(–6.7Tru.Hcrtr2:GAL4-VP16);Tg(UAS:E1b-Kaede)*. Dashed lines outline the projection nucleus. Cell and transcript expression outside the projection nucleus is removed for visual clarity. Images taken at 5 dpf in sagittal mount. All scale bars, 20 μm.

neurons remaining (**Figure 4D**). We conclude that projection neurons remain competent to assemble with spatially-appropriate targets.

If motor neurons are required to initiate synaptogenesis, then projection neurons should fail to develop pre-synaptic machinery. To test this hypothesis, we performed fluorescent in situ hybridization against common pre-synaptic transcripts: synaptophysin a (*sypa*), synaptic vesicle glycoprotein (*sv2*), and synapsin I (*syn1*). In both *phox2a* mutants and controls, we observed robust transcript expression in projection neuron somata at 5 dpf (**Figure 4E**), well-after synaptogenesis onset in wildtype larvae (**Goldblatt et al., 2023**). Motor partner-derived signals are thus not required for projection neurons to develop the necessary components for synaptogenesis.

Although motor neurons may play later roles in selecting and/or refining pre-motor input specificity, our data supports that projection neurons remain anatomically and molecularly poised to assemble

with appropriate targets. We predict that absent collaterals and synapses reflect a lack of adhesive contact necessary to stabilize (*Dalva et al., 2007*, *Moreland and Poulain, 2022*), but not instruct the formation of nascent structures.

## The transcriptional profiles of projection neurons are intact in the absence of motor partners

We found that functional and anatomical connectivity, from peripheral sensors to motor targets, develop independently of motor partners. Fate in the vestibulo-ocular reflex circuit follows from connectivity (*Szentágothai, 1964*), but neuronal fate can also be defined with respect to unique transcriptional signatures. Previously, we developed a sequencing pipeline to discover transcription factors that specify functional subtypes of spinal motor neurons and evaluate the consequences of perturbations on transcriptional fate (*D'Elia et al., 2023*). We adapted this approach to determine if loss of motor-derived signals changed the transcriptional profiles of projection neurons.

We compared the transcriptional profiles of projection neurons in *phox2a* mutants and sibling controls (*Figure 5A*) using bulk RNA sequencing. Sibling controls included both wildtype and heterozygous *phox2a* larvae (*Figure 1—figure supplement 2*). We performed sequencing experiments at 72 hr post-fertilization (hpf), after projection neuron differentiation is complete and synaptogenesis to motor targets has peaked (*Goldblatt et al., 2023*). We sequenced projection neurons labeled by *Tg(–6.7Tru.Hcrtr2:GAL4-VP16);Tg(UAS-E1b:Kaede)* (*Scott et al., 2007*; *Lacoste et al., 2015*; *Bianco et al., 2012*; *Schoppik et al., 2017*; Materials and methods, *Figure 5B*, *Figure 5—figure supplement 1*). Neurons labeled in this line include, but are not exclusive to the projection neurons in the tangential nucleus used for vertical gaze stabilization. Therefore, we evaluated our bulk RNA sequencing dataset in the context of a single-cell reference atlas derived from the same transgenic line (Materials and methods, *Figure 5—figure supplement 2*) to minimize noise from other labeled populations. We used *evx2* (*Sugioka et al., 2023*) as a reference, as it was expressed in all projection neurons (*Figure 5—figure supplement 2*) and highly detected (36%) in singly-profiled projection neurons.

There were strikingly few differentially-expressed genes in projection neurons between *phox2a* siblings and null mutants (*Figure 5C*, *Table 3*). All candidate differentially-expressed genes were lowly-expressed (detected in <10% of reference projection neurons, *Figure 5D*). To determine if any candidates were differentially expressed in projection neurons, we used a fluorescent in situ hybridization method *Choi et al., 2018* in which fluorescence intensity correlates with detected transcript expression reliably across individual larvae (*Figure 5—figure supplement 3*). We evaluated eight candidate genes (*Figure 5E, F*); four with the highest detection levels in projection neurons (*satb1a*, *rxfp2a*, *mapk6*, and *p4hb*), two with high significance and fold change thresholds (*itga9*, *twf1b*), and two highly-detected controls (*evx2*, *myt1la*). Qualitatively, we observed no differences in expression patterns between *phox2a* mutants and siblings.

We considered that our inability to detect differentially-expressed genes could arise from our exclusion of candidates based on their expression in our reference single-cell atlas. Therefore, we repeated our analyses in unfiltered bulk sequencing data. The top 50 highest-expressed genes in *phox2a* siblings were highly detected in singly-profiled neurons labeled by *Tg(–6.7Tru.Hcrtr2:GAL4-VP16);Tg(UAS-E1b:Kaede)*, including projection neurons (*Table 4*) (mean detection in projection neurons: 55%±21%, min = 0%, max = 87%). This suggests that our dissections adequately captured our target population. However, we again identified few differentially-expressed genes in our unfiltered data (*Figure 5—figure supplement 4A–C*), with substantial decreases as significance stringency increased. In situ hybridization validated that top candidates remained lowly expressed in projection neurons in both *phox2a* siblings and mutants (*Figure 5—figure supplement 4D*), Importantly, nearly all candidates had low detection across all neurons in our reference single-cell atlas and had predicted expression in populations such as glia and the caudal hindbrain (Materials and methods, *Table 5*). Notably, some candidates were highly expressed in a subset of medial vestibular nucleus neurons, which lies on the medial edge of the tangential nucleus and expressed *phox2a* (Materials and methods, *Figure 5—figure supplement 5*, and *Table 6*). Together, we conclude that any differential gene expression in our data either reflects noise or contamination from other labeled populations, but not projection neurons in the tangential nucleus.

We acknowledge the possibility that our in situ method is insufficiently quantitative to detect subtle differences in expression. Similarly, despite using both bulk and single-cell RNA sequencing

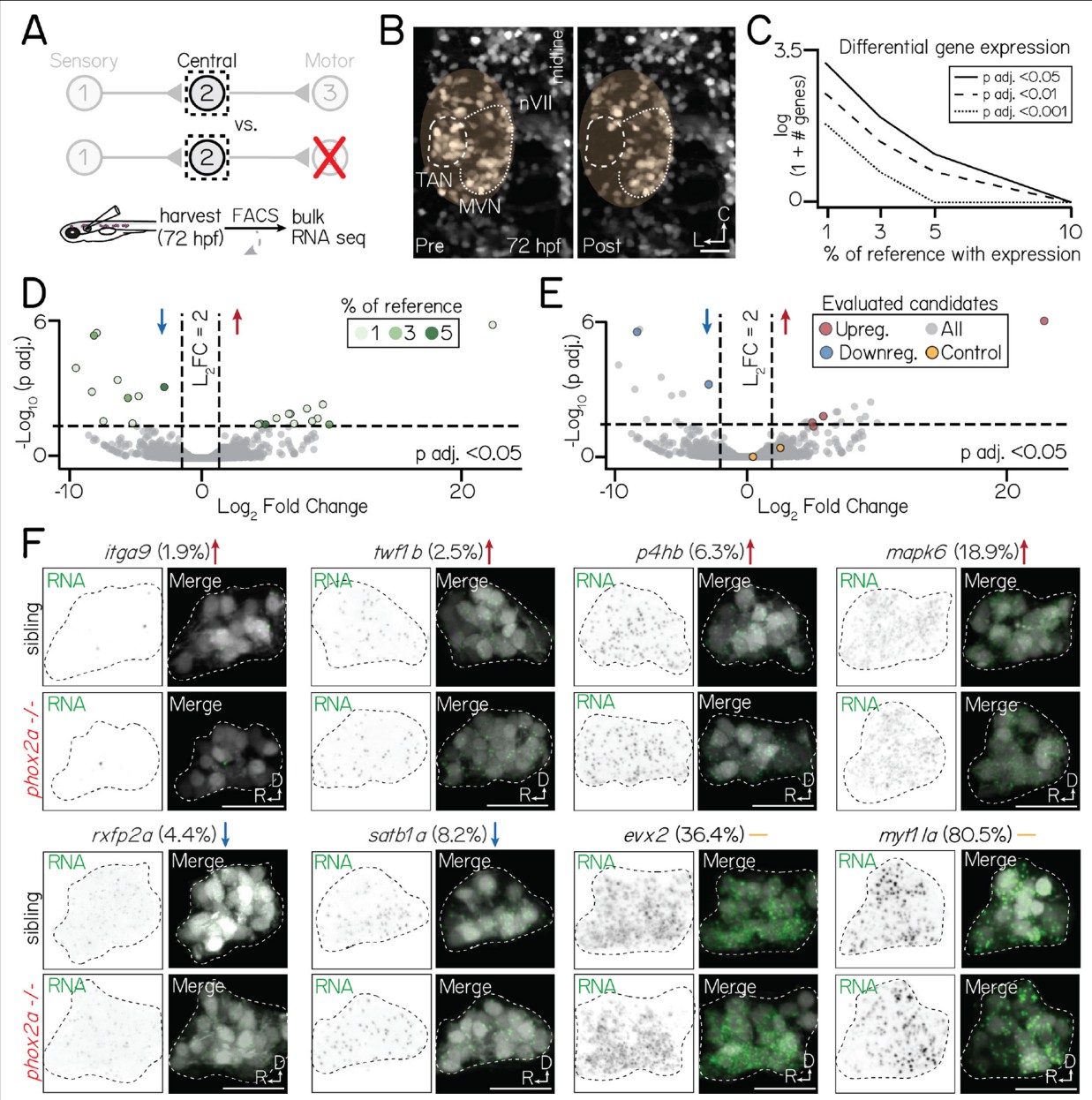

**Figure 5.** Motor neurons are dispensable for normal transcriptional profiles of projection neurons. Associated with *Figure 5—figure supplement 1*, *Figure 5—figure supplement 4*, *Table 3*. (**A**) Schematic of sequencing approach. Central projection neurons (*Tg(–6.7Tru.Hcrtr2:GAL4-VP16);Tg(UAS:E1b-Kaede)*) are harvested from 3 dpf larvae. Flow cytometry is used to exclude neurons not labelled by *Tg(–6.7Tru.Hcrtr2:GAL4-VP16)*. Bulk RNA sequencing is performed to compare the profiles of projection neurons in siblings and *phox2a* null mutants. (**B**) Example of projection neurons before (left) and after (right) harvesting. Neurons visualized with *Tg(isl1:GFP);Tg(–6.7Tru.Hcrtr2:GAL4-VP16);Tg(UAS:E1b-Kaede)*. Dashed lines outline projection neurons in the tangential nucleus; dotted lines, medial vestibular nucleus. Yellow region shows margin of harvesting error: non-projection neurons that may be included in bulk sequencing dataset. (**C**) Number of differentially expressed genes in projection neurons at 3 dpf after applying progressive filters based on gene expression in a reference single-cell dataset. Data shown on logarithmic scale. Solid, dashed, and dotted lines represent differentially-expressed gene with p adjusted<0.5, p adjusted<0.01, or p adjusted<0.001 significance, respectively. (**D**) Volcano plot showing differentially expressed genes in projection neurons between control and *phox2a* null larvae at 3 dpf. Dashed lines represent significance cutoffs: horizontal line, p >0.05; vertical line, $\log_2$ Fold Change >2.0. Each circle is a gene. Genes to the left and right of 0 on the horizontal axis show downregulated and upregulated genes, respectively. Colors indicate percent of reference cells that express a given gene. Grey-colored genes are below both significance thresholds. (**E**) Same data as Figure 5D. Colored genes show eight candidates evaluated with fluorescent in situ hybridization: red, upregulated; blue, downregulated; yellow, highly-expressed controls (*evx2*). (**F**) Fluorescent in situ hybridization against candidate genes that met projection neuron filter criteria. Top row shows sibling controls; bottom row, *phox2a* null mutants. For each gene, left panels show RNA probe (green) and right panels show merge with projection neurons labeled by *Tg(–6.7Tru.Hcrtr2:GAL4-VP16)* (grey). Dashed lines outline the projection nucleus. Cell

*Figure 5 continued on next page*

*Figure 5 continued*

and transcript expression outside the projection nucleus is masked for visual clarity. Arrows denote whether genes are upregulated (red), downregulated (blue), or not significantly changed (yellow). Percentage refers to fraction of cells in a single-cell RNA sequencing reference atlas (Materials and methods) with detected transcript. Candidates: *itga9* ($\log_2$ fold change = 23.0, p adj.=$3.9 \times 10^{-6}$), *twf1b* ($\log_2$ fold change = 5.9, p adj.=0.024), *p4hb* ($\log_2$ fold change = 5.1, p adj.=0.04), *mapk6* ($\log_2$ fold change = 5.1, p adj.=0.06), *rxfp2a* ($\log_2$ fold change = −8.5, p adj.=$1.1 \times 10^{-5}$), *satb1a* ($\log_2$ fold change = −3.0, p adj.=0.001), *evx2* ($\log_2$ fold change = 0.46, p adj.=0.99), *myt1la* ($\log_2$ fold change = 2.6, p adj.=0.44). All scale bars, 20 μm.

The online version of this article includes the following figure supplement(s) for figure 5:

**Figure supplement 1.** Flow cytometry gating strategy to sort fluorescently-labeled neurons for bulk RNA sequencing.

**Figure supplement 2.** Molecular identification of projection neurons using a reference single-cell RNA sequencing atlas.

**Figure supplement 3.** Visualization of transcripts in siblings and *phox2a* null mutants with fluorescent in situ hybridization is (1) consistent across larvae and (2) scales with predicted detection in projection neurons.

**Figure supplement 4.** Differential gene expression in an unfiltered bulk sequencing dataset of siblings and *phox2a* mutants.

**Figure supplement 5.** phox2a expression in the medial vestibular nucleus may underscore differential gene expression phenotypes in bulk data.

approaches, we may lack the resolution to uncover differential gene expression within projection neurons, particularly given the inclusion of heterozygous *phox2a* siblings in controls. Nevertheless, consistent with functional and anatomical characterization, our sequencing data argues that projection neurons acquire the correct transcriptional profiles in the absence of motor partner-derived signals. Our findings are reminiscent of recent reports that the molecular signatures of spinal interneurons develop independently of motor partners (*Sweeney et al., 2018*).

## Discussion

Here, we show that motor neurons are dispensable for fate specification in a canonical sensorimotor circuit. We first demonstrated that peripheral sensory and central projection neurons develop appropriate, directionally-selective connectivity and topography independently of their motor partners. Next, we established that projection neurons remain anatomically and molecularly competent to assemble with motor partners. Lastly, we show that loss of motor neurons does not meaningfully alter the transcriptional signatures of their pre-motor projection neuron partners. By providing causal evidence against an instructional role of motor partners for sensory connectivity, our work forces a revision of the current model for vestibulo-ocular reflex circuit formation. As proper connectivity across multiple synapses is foundational for proper function, our work speaks to mechanisms responsible for sensorimotor circuit assembly.

### Transcriptional influences on motor neuron fate specification

While the primary focus of our work was circuit assembly, we found that, unexpectedly, *phox2a* acts in a dose-dependent manner to specify extraocular motor pool fate. Key evidence comes from *phox2a* heterozygotes, in which the earliest-born dorsal neurons in nIII are lost but later-born neurons in nIII/nIV are intact. This observation extends prior characterizations of *phox2a* mutations in zebrafish (*Guo et al., 1999*), chick (*Hasan et al., 2010*), and human (*Nakano et al., 2001*, *Bosley et al., 2006*). Prior work hypothesized that *phox2a* dosage may regulate midbrain motor neuron differentiation into visceral and somatic types (*Hasan et al., 2010*). In other systems, transcription (*DeKoter and Singh, 2000*, *Liu et al., 2007*, *Sansom et al., 2009*), growth (*Vemaraju et al., 2012*) and axon guidance factors (*Komiyama et al., 2007*) can act in such a graded manner to regulate coarse cell type specification and wiring specificity. We extend these ideas to show that *phox2a* dose-dependency acts both over closely-related subtypes (pools within a single cranial nucleus) and along a temporal axis, where partial dosage preferentially targets the earliest-born neurons (*Greaney et al., 2017*). Specifically, if *phox2a* is expressed in neural progenitor cells that give rise to nIII/nIV, then the earliest-born motor neurons would have the shortest exposure to *phox2a*.

Molecular insight into ocular motor neuron pool specification is sparse but would be welcome given the strong links between genetic development and ocular motor disease (*Bosley et al., 2006*, *Park et al., 2016*, *Cheng et al., 2014*, *Cederquist et al., 2012*). For example, subpopulation markers could resolve the topography of pools within dorsal nIII; whether IR/MR pools are spatially segregated or intermingled (*Evinger et al., 1987*, *Greaney et al., 2017*); whether the medial/lateral axis

**Table 3.** Differentially expressed genes in projection neurons.
Star indicates a gene was evaluated using fluorescent in situ hybridization. # symbol indicates a gene was also differentially expressed in adjacent *phox2a*-expressing medial vestibular neurons (see *Figure 5—figure supplement 5*). "% of projection neurons with expression" refers to detection in a filtered subset of projection neurons from a single-cell reference atlas of neurons labeled in *Tg(–6.7Tru.Hcrtr2:GAL4-VP16);Tg(UAS-E1b:Kaede)* (Materials and methods, *Figure 5—figure supplement 2*). Genes sorted by p adjusted value. Data associated with *Figure 5*.

| Gene | % of projection neurons with expression | Log$_2$ fold change | p adjusted |
|---|---|---|---|
| Upregulated | | | |
| * # itga9 | 1.9 | 23.0 | 3.9E-06 |
| si:dkey-54n8.2 | 1.3 | 9.6 | 0.007 |
| myof | 1.3 | –8.3 | 0.010 |
| gbe1a | 1.9 | 7.0 | 0.01 |
| # dysf | 1.9 | 6.9 | 0.016 |
| * # twf1b | 2.5 | 5.9 | 0.024 |
| cers3a | 2.5 | 9.2 | 0.025 |
| asip2b | 1.3 | 8.7 | 0.032 |
| # pole | 1.3 | 7.3 | 0.041 |
| abtb2a | 3.1 | 4.7 | 0.041 |
| * # p4hb | 6.3 | 5.1 | 0.044 |
| postnb | 3.8 | 10.1 | 0.044 |
| fhdc3 | 2.5 | 4.5 | 0.044 |
| Downregulated | | | |
| msmo1 | 1.3 | –8.3 | 8.4E-06 |
| *rxfp2a | 4.4 | –8.5 | 1.1E-05 |
| si:ch73-204p21.2 | 1.3 | –9.9 | 2.2E-04 |
| tsta3 | 0.0 | –6.6 | 6.8E-04 |
| * # satb1a | 8.2 | –3.0 | 0.001 |
| # polrmt | 1.9 | –8.7 | 0.002 |
| # znf975 | 1.3 | –5.0 | 0.003 |
| # phldb1a | 3.1 | –5.8 | 0.004 |
| asns | 2.5 | –7.8 | 0.032 |
| nr1i2 | 1.9 | –5.5 | 0.039 |
| Control | | | |
| evx2 | 36.4 | 0.46 | 0.99 |
| myt1la | 80.5 | 2.6 | 0.44 |

reflects functional differences among motor neuron subtypes; and whether/how local interactions between motor neuron pools contributes to fate specification (*Knüfer et al., 2020*). In spinal circuits, the rich molecular understanding of motor pool specification (*Lin et al., 1998*, *Dasen et al., 2005*, *Dasen et al., 2008*, *Hanley et al., 2016*, *D'Elia et al., 2023*) has enabled targeted perturbations of pool identity, allowing for major discoveries of their roles in circuit assembly (*Vrieseling and Arber, 2006*, *Sürmeli et al., 2011*, *Sweeney et al., 2018*, *Philippidou and Dasen, 2013*). Our findings thus represent a step forward towards understanding how developmental deficits may contribute to ocular motor disorders (*Whitman and Engle, 2022*).

**Table 4.** Top 50 expressed genes in an unfiltered bulk RNA sequencing dataset of *phox2a* siblings. '% of unfiltered 10 x neurons' refers to gene detection in a single-cell atlas of neurons labeled in *Tg(–6.7Tru.Hcrtr2:GAL4-VP16);Tg(UAS-E1b:Kaede)* (n=1,468 neurons). '% of projection neurons' refers to gene detection in a subset of the single-cell atlas containing projection neurons in the tangential nucleus (n=159 neurons). Data associated with *Figure 5*.

| Gene | % of unfiltered 10 x neurons with expression | % of projection neurons |
|---|---|---|
| *ints5* | 23.2 | 15.1 |
| *stmn1b* | 78.6 | 78.6 |
| *sox4a* | 6.5 | 3.1 |
| *basp1* | 61.4 | 62.9 |
| *hmgb3a* | 68.2 | 69.8 |
| *ptmaa* | 84.5 | 77.4 |
| *gapdhs* | 28.9 | 30.2 |
| *pnrc2* | 81.3 | 78.6 |
| *snap25a* | 65.7 | 78.0 |
| *gpm6ab* | 81.1 | 78.6 |
| *calm3b* | 0.0 | 0.0 |
| *marcksl1b* | 88.8 | 86.2 |
| *tuba1c* | 59.9 | 54.7 |
| *cd81a* | 43.3 | 49.7 |
| *meis1b* | 87.9 | 87.4 |
| *rtn1a* | 73.4 | 74.8 |
| *elavl3* | 87.1 | 82.4 |
| *hmgb1b* | 57.2 | 49.7 |
| *ptmab* | 80.7 | 73.6 |
| *zc4h2* | 56.9 | 58.5 |
| *meis2b* | 57.1 | 49.7 |
| *slc25a5* | 51.4 | 55.3 |
| *mab21l2* | 62.7 | 68.6 |
| *h3f3c* | 69.1 | 61.6 |
| *rtn1b* | 36.4 | 33.3 |
| *elavl4* | 78.7 | 67.3 |
| *gng3* | 37.2 | 42.8 |
| *pik3r3b* | 77.4 | 83.6 |
| *tubb5* | 25.3 | 25.2 |
| *histh1l* | 61.0 | 62.9 |
| *serinc1* | 51.9 | 59.1 |
| *ckbb* | 23.5 | 30.8 |
| az1a | 43.5 | 49.7 |
| az1b | 36.9 | 38.4 |
| actb1 | 23.6 | 28.3 |
| ywhaba | 36.2 | 40.9 |

*Table 4 continued on next page*

*Table 4 continued*

| Gene | % of unfiltered 10 x neurons with expression | % of projection neurons |
|------|---------------------------------------------|-------------------------|
| *ywhag2* | 36.2 | 50.9 |
| *si:ch211-222l21.1* | 73.8 | 63.5 |
| *si:dkey-276j7.1* | 45.8 | 56.0 |
| *aldocb* | 19.3 | 17.6 |
| *actb2* | 27.0 | 30.8 |
| *tmem59l* | 39.8 | 56.6 |
| *calm2b* | 37.9 | 47.2 |
| *hmgn6* | 73.6 | 65.4 |
| *h2afx1* | 59.6 | 53.5 |
| *cd99l2* | 32.4 | 36.5 |
| *cirbpb* | 77.8 | 73.6 |
| *ppdpfb* | 74.5 | 65.4 |
| *stxbp1a* | 52.3 | 66.0 |
| Control | | |
| *evx2* | 33.8 | 36.4 |

## Motor neurons: active or passive architects of pre-motor connectivity?

Our discoveries advance outstanding controversies over whether motor neurons actively or passively shape pre-motor connectivity. We find that extraocular motor neuron axons do not serve as 'pioneers' (**Supèr et al., 1998**, **McConnell et al., 1989**, **Bentley and Keshishian, 1982**, **Bate, 1976**, **Pike et al., 1992**), with pre-motor axon targeting following passively from motor-derived pathfinding signals (**Matise and Lance-Jones, 1996**). Such a model predicts that projection neuron targeting would be entirely ablated after constitutive loss of extraocular motor neurons and their secreted signals (**Landmesser and Honig, 1986**, **Scott, 1988**, **Swanson and Lewis, 1986**, **Tosney and Hageman, 1989**, **Whitlock and Westerfield, 1998**). Instead, we observed that projection neurons still establish long-range (hindbrain to midbrain) axonal projections, with appropriate spatial segregation that matches the topography of their motor partners (**Greaney et al., 2017**, **Liu et al., 2022**, **Goldblatt et al., 2023**). Our findings complement reports in spinal circuits that pre-motor targeting is grossly appropriate after manipulating the spatial source of, but not ablating, potential pathfinding signals (**Sürmeli et al., 2011**), and that the transcriptional fate of pre-motor projection neurons similarly develops independently (**Sweeney et al., 2018**). We point to the late development of ocular musculature (**Easter and Nicola, 1997**, **Noden et al., 1999**) compared to spinal musculature (**Matise and Lance-Jones, 1996**) as a potential source of the dispensability of muscle-derived signals.

Our work is also inconsistent with the strongest form of the "retrograde" hypothesis for vestibulo-ocular reflex circuit assembly. Originally, the retrograde model posited that motor neurons release a diffusable or cell-surface available signal that instructs pre-motor collaterals to sprout and then innervate specific pools, enabling behavioral specificity (**Glover, 2003**, **Straka, 2010**). Here, the proper spatial and temporal segregation of projection neuron axons suggests they remain poised to wire with spatially-appropriate (dorsal/ventral pools) targets. Additional evidence comes from incomplete *phox2a* knockouts (1–5% of nIII/nIV remaining), where projection neurons still form collaterals, though not robustly or reliably. We predict that projection neuron axons do not require a target-derived cue to grow, search, and synapse onto motor targets, and simply lack the adhesive contact necessary to stabilize nascent structures (reviewed in **Dalva et al., 2007**; **Moreland and Poulain, 2022**).

Nevertheless, extraocular motor neurons might still play an active or passive role in selecting and/or refining input specificity from their projection neuron partners. In spinal circuits, motor pool position passively imposes geometric constrains on pre-motor axon targeting (**Sürmeli et al., 2011**, **Bikoff et al., 2016**), and manipulating the dendritic structure of motor neuron axons transforms input

**Table 5.** Top 50 differentially expressed genes in an unfiltered bulk RNA sequencing dataset of *phox2a* siblings and null mutants.

One star indicates a gene was retained in a filtered subset of projection neurons; %, evaluated using fluorescent in situ hybridization. '% of unfiltered 10 x neurons' refers to gene detection in an unfiltered single-cell reference atlas of neurons labeled in *Tg(–6.7Tru.Hcrtr2:GAL4-VP16);Tg(UAS-E1b:Kaede)* (n=1,468 neurons). Putative origin inferred from gene expression in the annotated 10 x dataset (Materials and methods, *Figure 5—figure supplement 2*). Genes sorted by p adjusted value. Data associated with *Figure 5*.

| Gene | % of unfiltered 10 x neurons with expression | Putative origin | Log$_2$ fold change | p adjusted |
|---|---|---|---|---|
| **Upregulated** | | | | |
| *macc1* | 0.1 | r4-6 | 24.0 | 3.2E-06 |
| *CR559941.1* | 0.0 | | 23.7 | 3.4E-06 |
| *si:dkey-65b12.6* | 0.0 | | 23.5 | 3.4E-06 |
| *si:ch73-106n3.2* | 0.1 | | 23.5 | 3.4E-06 |
| *mcm10* | 0.1 | MNs | 23.4 | 3.4E-06 |
| *si:ch211-244o22.2* | 0.5 | r4-6 | 23.4 | 3.4E-06 |
| *dre-mir-10a* | 0.0 | | 23.3 | 3.5E-06 |
| *itga4* | 0.3 | r5-6 (inhibitory) | 23.2 | 3.5E-06 |
| *si:dkeyp-87d8.8* | 0.0 | | 23.2 | 3.6E-06 |
| *arsj* | 0.5 | MNs | 23.0 | 3.9E-06 |
| *tlr1* | 0.0 | | 23.0 | 3.9E-06 |
| * % *itga9* | 2.3 | r4-7 | 23.0 | 3.9E-06 |
| *tofb* | 0.5 | r4-6 | 9.1 | 1.4E-05 |
| *myo7ba* | 0.4 | r4-7 | 9.8 | 1.0E-04 |
| *zfand1* | 0.1 | MNs | 9.0 | 1.5E-04 |
| % *slc22a7a* | 0.7 | r4-7 | 10.2 | 1.5E-04 |
| *agrp* | 0.0 | | 13.4 | 4.1E-04 |
| *si:dkey-46i9.6* | 0.1 | r5-7 | 7.7 | 6.8E-04 |
| *muc2.2* | 0.0 | | 9.4 | 6.9E-04 |
| *cd37* | 0.0 | | 9.1 | 9.8E-04 |
| *musk* | 0.3 | r4-6 | 9.4 | 1.2E-03 |
| *mcamb* | 0.2 | r5-7 | 8.3 | 2.7E-03 |
| *ppp1r42* | 0.5 | r5-6 (inhibitory) | 7.9 | 3.1E-03 |
| *CR677513.1* | 0.0 | | 9.9 | 3.5E-03 |
| **Downregulated** | | | | |
| * % *satb1a* | 7.9 | r4-7 (inc inhib), MNs | –8.6 | 1.0E-06 |
| *znf975* | 0.7 | r4-6 | –8.3 | 1.5E-06 |
| *phldb1a* | 0.6 | r5-7 (inc inhib) | –9.2 | 1.5E-06 |
| *TSTA3* | 0.0 | | –9.7 | 3.4E-06 |
| *si:dkey-24p1.6* | 0.0 | | –8.3 | 8.4E-06 |
| *si:dkey-77f5.14* | 0.2 | r5-7 | –8.5 | 1.1E-05 |

*Table 5 continued*

| Gene | % of unfiltered 10 x neurons with expression | Putative origin | Log₂ fold change | p adjusted |
|---|---|---|---|---|
| tha1 | 0.1 | MVN | –10.3 | 2.1E-05 |
| serpinh2 | 0.5 | r4-6 | –9.0 | 3.7E-05 |
| ghrh | 0.3 | r4-7 | –9.5 | 6.9E-05 |
| asah1b | 0.8 | r4-7 | –7.8 | 9.9E-05 |
| msmo1 | 0.9 | r5-7, inc inhib | –8.9 | 1.1E-04 |
| tagln2 | 0.3 | glia | –8.4 | 2.2E-04 |
| zgc:174863 | 0.1 | MNs | –9.9 | 2.2E-04 |
| * % rxfp2a | 3.2 | r4-7, inc inhib | –6.6 | 6.8E-04 |
| bmp4 | 0.7 | r4-7 | –6.8 | 6.8E-04 |
| cfl1l | 0.1 | r4-6 | –8.4 | 6.8E-04 |
| * polrmt | 4.2 | r4-7, inc inhib | –8.8 | 6.9E-04 |
| anxa2a | 0.6 | r4-7 | –3.0 | 1.3E-03 |
| galr1a | 0.3 | MVN | –9.1 | 1.4E-03 |
| selenow2b | 0.1 | | –8.0 | 1.8E-03 |
| % bckdhbl | 1.4 | r4-7, glia, MNs | –8.7 | 2.1E-03 |
| boka | 0.5 | r5-7 | –8.6 | 2.9E-03 |
| cyldb | 0.2 | r4-7 | –7.9 | 3.0E-03 |
| pon2 | 0.6 | r4-7, glia, MNs | –5.0 | 3.1E-03 |
| si:ch73-204p21.2 | 0.3 | r5-7, inc inhib | –8.2 | 3.5E-03 |
| and2 | 0.1 | r4-6 | –5.8 | 3.7E-03 |
| Control | | | | |
| evx2 | 33.8 | r4-7 | 0.46 | 0.99 |

specificity (*Vrieseling and Arber, 2006*, *Balaskas et al., 2019*). Genetic perturbations of nIII/nIV motor neuron position selectively compromise ocular responses to directional visual stimuli (*Knüfer et al., 2020*), though the circuit-level origin of such impairments is unclear. For the vestibulo-ocular reflex circuit, transforming all motor pools to the same fate or genetically "scrambling" pool position could resolve whether motor input specificity is truly hard-wired in projection neurons, or whether projection neurons instead target gross spatial domains irrespective of partner identity (*Sürmeli et al., 2011*). Motor neuron-derived signals are of course capable of shaping their input by strengthening/weakening their inputs. For example, motor neurons may play separate roles in recruiting their premotor partners into functional ensembles and synchronizing their spontaneous activity (*O'Donovan et al., 1994*, *Wenner and O'Donovan, 2001*, *Warp et al., 2012*, *Song et al., 2016*, *Wan et al., 2019*, *Younes et al., 2024*). Importantly, our results suggest that such signals will not define the fate of projection neurons, and by extension, circuit architecture.

We note that our study does not eliminate one additional source of post-synaptic partner signals to projection neurons. As in primates (*McCrea et al., 1987*), projection neurons also contact neurons in the interstitial nucleus of Cajal, also known as the nucleus of the medial longitudinal fasciculus (INC/nMLF; *Bianco et al., 2012*). INC/nMLF neurons project early in development (*Mendelson, 1986*) to spinal circuits used for postural stabilization during swimming (*Severi et al., 2014*, *Thiele et al., 2014*, *Wang and McLean, 2014*, *Tanimoto et al., 2022*, *Berg et al., 2023*). Notably, ablation of projection neurons disrupts postural stability (*Sugioka et al., 2023*). As we did not observe postural deficits in *phox2a* mutants, we infer that projection neuron connectivity to INC/nMLF targets is present and functional. Correspondingly, the development of projection neuron collaterals and synapses to INC/

**Table 6.** Differentially expressed genes in *phox2a*-expressing medial vestibular neurons.
Star indicates a gene was evaluated in projection neurons using fluorescent in situ hybridization.
# indicates a gene was significantly differentially expressed in projection neurons. '% of medial vestibular neurons' refers to detection in a subset of *phox2a*-expressing medial vestibular neurons in a single-cell reference atlas (Materials and methods, *Figure 5—figure supplement 2*). '% of projection neurons with expression' refers to detection in a filtered subset of projection neurons. Gene sorted by p adjusted value. Data associated with *Figure 5*.

| Gene | % of medial vestibular neurons | % of projection neurons | Log$_2$ fold change | p adjusted |
|---|---|---|---|---|
| Upregulated | | | | |
| *itga4* | 2.2 | 0.6 | 23.2 | 3.5E-06 |
| * # *itga9* | 6.7 | 1.9 | 23.0 | 3.9E-06 |
| *musk* | 2.2 | 0.6 | 9.4 | 0.001 |
| # *dysf* | 6.7 | 1.9 | 6.9 | 0.016 |
| * # *twf1b* | 8.9 | 1.9 | 5.9 | 0.024 |
| *gabrr2a* | 2.2 | 2.5 | 6.8 | 0.040 |
| # *pole* | 2.2 | 0.0 | 7.3 | 0.041 |
| * # *p4hb* | 15.6 | 1.3 | 5.1 | 0.044 |
| *col27a1b* | 2.2 | 0.6 | 8.0 | 0.044 |
| Downregulated | | | | |
| *asah1b* | 2.2 | 0.0 | −8.3 | 1.5E-06 |
| *boka* | 2.2 | 0.0 | −9.2 | 1.5E-06 |
| * # *satb1a* | 6.7 | 8.2 | −3.0 | 0.001 |
| # *polrmt* | 4.4 | 1.9 | −8.7 | 0.002 |
| # *znf975* | 2.2 | 1.3 | −5.0 | 0.003 |
| # *phldb1a* | 2.2 | 3.1 | −5.8 | 0.004 |
| *fosl1a* | 2.2 | 0.0 | −5.4 | 0.016 |
| *pitpnaa* | 2.2 | 0.0 | −8.8 | 0.036 |
| *sgpp1* | 2.2 | 0.6 | −7.1 | 0.038 |

nMLF neurons appeared qualitatively normal in *phox2a* mutants, supporting our interpretation that projection neurons retain the capacity to properly assemble with post-synaptic targets even though similar structures to extraocular motor neurons are absent. In the future, if a similarly specific marker like *phox2a* is identified that labels the INC/nMLF, it will be possible to test whether these neurons play a role in vestibulo-ocular reflex circuit development.

## Alternative mechanisms for fate specification and sensory input specificity in projection neurons

What is the origin of signals that govern projection neuron fate and sensory input specificity, if not motor-derived? In comparable systems, fate signals can be intrinsically-expressed or originate from extrinsic sources. For example, intrinsic genetic mechanisms assemble laminar connectivity in visual circuits (*Yamagata et al., 2002*, *Yamagata and Sanes, 2008*) and facilitate sensorimotor matching in spinal circuits (*Arber et al., 2000*, *Vrieseling and Arber, 2006*, *Pecho-Vrieseling et al., 2009*, *Bonanomi and Pfaff, 2010*). In directionally-selective retinal circuits, subtype fate is established in a similar manner (*Duan et al., 2014*, *Al Khindi et al., 2022*). In 'intrinsic' models, synaptic specificity arises from molecular matching between subtypes (*Kurmangaliyev et al., 2019*; *Yoo et al., 2023*; *Dasen, 2009*). Alternatively, in somatosensory and auditory circuits, transcriptional fate depends on extrinsic signals such as growth factors (*Sharma et al., 2020*) and sensation (*Shrestha et al., 2018*),

respectively. In spinal circuits, positional fate, which constrains connectivity (*Sürmeli et al., 2011*), is established by extrinsic codes such as morphogen gradients in early development (*Alaynick et al., 2011*) and Hox factors (*Dasen et al., 2005*, *Dasen et al., 2008*). In 'extrinsic' models, early inputs are often erroneous and refined by activity (*Tessier-Lavigne and Goodman, 1996*, *Goodman and Shatz, 1993*) or molecular factors (*Spead et al., 2023*). Collectively, these findings offer two alternative models for how vestibulo-ocular reflex circuit assembly emerges.

The tight links between birth order, somatic position, and stimulus selectivity (*Greaney et al., 2017*, *Goldblatt et al., 2023*, *Liu et al., 2022*, *Tanimoto et al., 2022*, *Zecca et al., 2015*, *Dyballa et al., 2017*) across vestibulo-ocular reflex circuit populations support an 'intrinsic' determination model. Further, neurogenesis and initial axon targeting develops contemporaneously for sensory afferents (*Fritzsch and Nichols, 1993*, *Altman and Bayer, 1982*, *Vemaraju et al., 2012*, *Zecca et al., 2015*), projection neurons (*McConnell and Sechrist, 1980*, *Maklad and Fritzsch, 2003*, *Auclair et al., 1999*, *Glover, 2003*, *Goldblatt et al., 2023*), and extraocular motor neurons (*Greaney et al., 2017*, *Altman and Bayer, 1981*, *Glover, 2003*, *Shaw and Alley, 1981*, *Puelles and Privat, 1977*, *Varela-Echavarría et al., 1996*), suggesting that neurons are poised to assemble with targets as early as their time of differentiation. Importantly, an 'intrinsic specification' model makes a testable prediction about how and when sensory selectivity should emerge across the circuit: projection neurons and extraocular motor neurons should be directionally selective as soon as pre-synaptic input is established. Such evidence would justify future molecular inquiries into the underlying genetic factors, expanding early characterizations of the mechanisms that shape hindbrain topography (*Moens and Prince, 2002*, *Kinkhabwala et al., 2011*), recent molecular profiling of the zebrafish hindbrain (*Farnsworth et al., 2020*, *Tambalo et al., 2020*), and reports of molecular matching between extraocular motor neurons and muscle (*Ferrario et al., 2012*, *Guthrie, 2007*). Operationally, the present study lays a foundation for molecular explorations of projection neuron subtype determinants by establishing bulk- and single-cell transcriptomic profiling and in situ validation pipelines.

Conversely, evidence that stimulus selectivity emerges gradually would suggest that sensory afferents and/or projection neurons initially wire indiscriminately and that circuit connectivity is refined in time by extrinsic forces. Prior work in the vestibulo-ocular reflex circuit has proposed developmental roles for sensory-derived trophic factors (*Peusner and Morest, 1977*) and activity-dependent refinement (*Ronca et al., 2016*, *Riley and Moorman, 2000*), though sensory afferents develop typically in the absence of utricular input (*Roberts et al., 2017*) and ocular motor behavior does not depend on stimulus-driven activity (*Ulrich et al., 2014*). Here, an 'extrinsic' determination model would predict that connectivity is established by an anterogradely-transmitted signal – that is, from sensory afferents to ocular muscles. If so, then future investigations might constitutively ablate sensory afferents to eliminate activity-driven, diffusible, or cell-surface instructional signals, similar to the present study. The directional bias in opsin-evoked activity in projection neurons (*Schoppik et al., 2017*), together with their transcriptional profiles established here, offer a clear readout of the role of sensory-derived factors. However, genetic targets exclusive to vestibular sensory afferents for gaze stabilization have not been identified, and tissue-specific genetic ablations remain limited in zebrafish. Looking ahead, resolving when and how stimulus selectivity emerges across the vestibulo-ocular reflex circuit will be key to understanding whether connectivity with pre- and/or post-synaptic partners instructs subtype fate, or whether subtype fate instructs connectivity.

## Conclusion

Here, we discovered that motor partners do not determine pre-motor fate and sensory connectivity for the projection neurons that stabilize gaze. Our results overturn the current model that stimulus selectivity and connectivity are retrogradely specified, a major step towards understanding the origin, and eventually nature, of mechanisms that assemble an archetypal sensorimotor reflex circuit. Instead, our data support and extend recent models in spinal systems that motor partners do not actively construct sensory-to-interneuron reflex circuit architecture, but may later refine their inputs. By defining the contribution of motor neurons to specification and sensory connectivity of gaze-stabilizing central projection neurons, our work speaks to principles of sensorimotor circuit assembly.

# Materials and methods

**Key resources table**

| Reagent type (species) or resource | Designation | Source or reference | Identifiers | Additional information |
|---|---|---|---|---|
| Chemical compound, drug | Tween | Thermo Fisher Scientific | BP337-100 | |
| Chemical compound, drug | 32% paraformaldehyde | Electron Microscopy Sciences | 15714 | |
| Chemical compound, drug | Proteinase K | Thermo Fisher Scientific | 25530049 | |
| Peptide, recombinant protein | Papain | Worthington Biochemical | LK003178 | |
| Chemical compound, drug | Hanks Buffered Salt Solution (HBSS) | Thermo Fisher Scientific | 14170112 | |
| Chemical compound, drug | Earl's Buffered Salt Solution (EBSS) | Thermo Fisher Scientific | 24010043 | |
| Peptide, recombinant protein | DNAse | Worthington Biochemical | LK003172 | |
| Peptide, recombinant protein | DAPI | Invitrogen | D1306 | |
| Chemical compound, drug | L15 Medium | Thermo Fisher Scientific | 11415064 | |
| Chemical compound, drug | Fetal bovine serum, qualified, triple-filtered | Thermo Fisher Scientific | A3160501 | |
| Peptide, recombinant protein | Collagenase Type 1 A | Sigma Aldrich | C9891-500MG | |
| Chemical compound, drug | Low melting point agarose | Thermo Fisher Scientific | 16520 | |
| Chemical compound, drug | Ethyl-3-aminobenzoic acid ethyl ester (MESAB) | Sigma Aldrich | E10521 | |
| Chemical compound, drug | Pancuronium bromide | Sigma Aldrich | P1918 | |
| Commercial assay, kit | in situ hybridization chain reaction v3.0 (HCR) | Molecular Instruments | N/A | |
| Commercial assay, kit | RNAqueous Micro Total RNA Isolation Kit | Thermo Fisher Scientific | AM1931 | |
| Commercial assay, kit | MEGAshortscript T7 Transcription Kit | Thermo Fisher Scientific | AM1354 | |
| Commercial assay, kit | QiaQUICK PCR Purification Kit | Qiagen | 28104 | |
| Commercial assay, kit | EnGen Spy Cas9 NLS | New England Biolabs | M0646T | |
| Strain, strain background (*Danio rerio*) | *Tg(–6.7Tru.Hcrtr2:GAL4-VP16)* | **Lacoste et al., 2015**; **Schoppik et al., 2017** | ZFIN: ZDB-TGCONSTRCT-151028–8 | |
| Strain, strain background (*Danio rerio*) | *Tg(UAS-E1b:Kaede)* | **Scott et al., 2007** | ZFIN: ZDB-TGCONSTRCT-070314–1 | |
| Strain, strain background (*Danio rerio*) | *Tg(isl1:GFP)* | **Higashijima et al., 2000** | ZFIN: ZDB-ALT-030919–2 | |
| Strain, strain background (*Danio rerio*) | *Tg(UAS:GCaMP6s)* | **Chen et al., 2013** | ZFIN: ZDB-TGCONSTRCT-140811–3 | |
| Strain, strain background (*Danio rerio*) | *phox2a^{d22}* | This study | N/A | |
| Strain, strain background (*Danio rerio*) | *phox2a^{d19}* | This study | N/A | |
| Strain, strain background (*Danio rerio*) | *phox2a^{i2}* | This study | N/A | |

*Continued on next page*

*Continued*

| Reagent type (species) or resource | Designation | Source or reference | Identifiers | Additional information |
|---|---|---|---|---|
| Sequence-based reagent (primers) | *phox2a* forward primer | Sigma Aldrich | N/A | CAGCCAGAGCAACGGCTTCC |
| Sequence-based reagent (primers) | *phox2a* reverse primer | Sigma Aldrich | N/A | AAGCCGACAACAGTGTGTGTGTAA |
| Sequence-based reagent (primers) | *phox2a* guide 1 | Sigma Aldrich | N/A | CTCGCCACCGCCAGCTGCAC |
| Sequence-based reagent (primers) | *phox2a* guide 2 | Sigma Aldrich | N/A | CTCCGGCTTCAGCTCCGGCC |
| Sequence-based reagent (oligonucleotides) | HCR probes | Integrated DNA Technologies | N/A | |
| Software, algorithm | Fiji/ImageJ | *Schindelin et al., 2012* | RRID: SCR_02285 | |
| Software, algorithm | Adobe Illustrator (2021) | Adobe | RRID: SCR_010279 | |
| Software, algorithm | Matlab 2020b | Mathworks | RRID: SCR_001622 | |
| Software, algorithm | Seurat v4 | *Hao et al., 2021* | https://satijalab.org/seurat | |
| Software, algorithm | CRISPR Guide RNA Design Tool | Benchling | https://benchling.com/crispr | |
| Other | 20 micron cell strainer | pluriSelect | 431002060 | Method details (Neuron harvesting) |
| Other | SH800z 100 micron sorting chip | Sony | LE-C3210 | Method details (Flow cytometry) |

## Resource availability

### Lead contact

Further information and requests for resources and reagents should be directed to and will be fulfilled by the lead contact, David Schoppik (schoppik@gmail.com).

### Materials availability

Mutant fish lines generated in this study will be deposited to the Zebrafish International Resource Center (ZIRC).

## Experimental model and subject details

### Fish care

All protocols and procedures involving zebrafish were approved by the New York University Langone School of Medicine Institutional Animal Care & Use Committee (IACUC; approval number IA16-00561). All larvae were raised at 28.5 °C at a density of 20–50 larvae in 25–40 ml of buffered E3 (1 mM HEPES added). Larvae used for photofill experiments were raised in constant darkness; all other fish were raised on a standard 14/10 h light/dark cycle. Larvae for experiments were between 3–5 days post-fertilization (dpf).

### Transgenic lines

Experiments were conducted on the *mifta*[-/-] background to remove pigment. All experiments used larvae from the F3 generation or older of a newly-created line of *phox2a* mutants (described below) on the following backgrounds: *Tg(isl1:GFP)* (*Higashijima et al., 2000*) to validate *phox2a* loss-of-function; *Tg(isl1:GFP);Tg(−6.7Tru.Hcrtr2:GAL4-VP16)* (*Lacoste et al., 2015*; *Schoppik et al., 2017*) to drive UAS reporter expression; *Tg(UAS-E1b:Kaede)* (*Scott et al., 2007*) for anatomical imaging experiments; and *Tg(UAS:GCaMP6s)* (*Thiele et al., 2014*) for calcium imaging experiments. All larvae were selected for brightness of fluorescence relative to siblings. Mendelian ratios were observed, supporting that selected larvae were homozygous for fluorescent reporter alleles.

## Generation of *phox2a* mutants

*phox2a* mutant lines were generated using CRISPR/Cas9 mutagenesis. Two guide RNAs (gRNAs) were designed using the Benchling CRISPR Guide RNA Design Tool (see: key resources table). gRNAs were located towards the 5' region of exon 1 to minimize the size of any translated protein. gRNAs were incubated with Cas9 protein before co-injection into *Tg(isl1:GFP)* embryos at the single cell stage. Injected embryos were screened for anatomical phenotypes (reduction in *isl1*-positive nIII/nIV motor neurons). Phenotypic embryos (F0) and their embryos were raised and genotyped via sequencing to identify and validate germline mutations. Three founders were identified and used for experiments: (1) *phox2a$^{d22}$* has a 22 bp deletion from base pairs 249–270, (2) *phox2a$^{d19}$* has a 19 bp deletion from base pairs 262–280, and (3) *phox2a$^{i2}$* has a 2 bp insertion (AG) from base pairs 261–262. Each mutation created a nonsense mutation, causing a predicted premature stop codon at the beginning of the homeobox. All alleles were validated using complementation assays, and larvae from all three alleles were used in experiments. For brevity, only one allele (*phox2a$^{d22}$*) is shown in *Figure 1*.

## Maintenance of *phox2a* adults

*phox2a* null larvae do not survive past 7 dpf. Sibling embryos (*phox2a$^{+/+}$* or *phox2a$^{+/-}$*) were raised and genotyped to identify heterozygotes for line propagation. Primers for genotyping are listed in the Key Resources table. Genomic DNA was amplified using a polymerase (DreamTaq PCR Master Mix 2 X, Thermo Fisher Scientific K1071), 60°annealing temperature, 30 second elongation time, and 35 cycles of PCR. PCR generates a 169 bp product (wildtype), 147 bp product (*phox2a$^{d22}$*), 150 bp product (*phox2a$^{d19}$*), or 171 bp product (*phox2a$^{i2}$*). *phox2a$^{d22}$* and *phox2a$^{d19}$* DNA was evaluated using gel electrophoresis; *phox2a$^{i2}$* was assessed via sequencing with the reverse primer (Genewiz, Azenta Life Sciences, South Plainfield, New Jersey).

## **Method details**

### Confocal imaging

Larvae were anesthetized in 0.2 mg/mL ethyl-3-aminobenzoic acid ethyl ester (MESAB, Sigma-Aldrich E10521, St. Louis, MO) prior to confocal imaging except where noted. Larvae were mounted dorsal side-up (axial view) or lateral side-up (sagittal view) in 2% low-melting point agarose (Thermo Fisher Scientific 16520) in E3. Images were collected on a Zeiss LSM800 confocal microscope with a 20 x water-immersion objective (Zeiss W Plan-Apochromat 20 x/1.0). Images of tangential nucleus soma and axons were acquired in a lateral mount with an 80x80 µm imaging window. Stacks spanned ~30–40 µm, sampled every 1 µm. Images of nIII/nIV motor neurons were acquired in a dorsal mount with a 213x106 µm imaging window; stacks spanned approximately 90 µm, sampled every 1.5 µm. Images to validate nIII/nIV expression in a lateral mount were acquired using a 319x319 µm imaging window. Raw image stacks were analyzed using Fiji/ImageJ (*Schindelin et al., 2012*).

### Identification of *phox2a* larvae

Prior to experiments, larvae were designated as *phox2a* mutants or sibling (wildtype/heterozygote) controls based on two criteria: gross loss of *Tg(isl1:GFP)* fluorescence in nIII/nIV at 2 dpf, visualized using a SugarCube LED Illuminator (Ushio America, Cypress CA) on a stereomicroscope (Leica Microsystems, Wetzlar, Germany) and absence of a swim bladder at 5 dpf. For anatomical and calcium imaging experiments, allele designations were validated using confocal imaging of nIII/nIV motor neurons: total or near-total loss of nIII/nIV neurons (null), selective loss of IR/MR neurons (heterozygote), or normal expression (wildtype). Designations were confirmed after experiments using genotyping. For RNA sequencing and fluorescent in situ experiments, sibling controls (wildtype/heterozygote) were combined.

### Birthdating of nIII/nIV motor neurons

Early-born neurons in nIII/nIV were optically tagged using in vivo birthdating (*Caron et al., 2008*, *Greaney et al., 2017*, *Goldblatt et al., 2023*) on *Tg(isl1:Kaede)$^{ch103}$* larvae (*Barsh et al., 2017*). Briefly, whole embryos were exposed to UV light for five minutes at experimenter-defined timepoints and subsequently raised in darkness to prevent background conversion. At 5 dpf, larvae were imaged on

a confocal microscope. Neurons born before the time of photoconversion expressed red, converted Kaede; neurons born after expressed only green, unconverted Kaede.

## Fluorescent in situ hybridization and imaging

Experiments were performed using Hybridization Chain Reaction (HCR) for whole-mount zebrafish larvae (*Choi et al., 2018*; *Ibarra-García-Padilla et al., 2021*). Probes were generated using the HCR 3.0 probe maker (*Kuehn et al., 2022*) using the sense sequence of the canonical gene cDNA from NCBI. All larvae were from the *Tg(isl1:GFP);Tg(–6.7Tru.Hcrtr2:GAL4-VP16);Tg(UAS-E1b:Kaede)* background. Larvae were pre-identified as null mutants or siblings (wildtype or heterozygotes) and combined in equal ratios (8–10 larvae per condition, 16–20 larvae total) into a single 5 mL centrifuge tube for fixation and HCR. Larvae were fixed overnight with 4% PFA in PBS at 4 ° C and stored in 100% methanol at –20 ° C. Subsequently, HCR was performed as described in *Ibarra-García-Padilla et al., 2021*, with adjustments to proteinase K incubation time based on age (2 dpf: 23 min incubation; 3 dpf: 30 min incubation; 5 dpf: 35 min incubation). HCR experiments used buffers and amplifiers from Molecular Instruments (Los Angeles, CA). DAPI staining was performed on some samples at 1:2000 from 5mg/ml stock solution, incubated overnight at 4 ° C. Samples were stored in 1 x PBS at 4 ° C and imaged on a confocal microscope within four days. Prior to imaging, larvae were re-screened for *Tg(isl1:GFP)* fluorescence to identify null mutants and sibling controls. For each probe, imaging parameters were determined using a sibling control and kept constant for all subsequent larvae. Comparable settings (within 1% laser power) were used across probes.

## Calcium imaging of tonic and impulse tilt stimuli responses

Experiments were performed as described in *Goldblatt et al., 2023* using Tilt-In-Place Microscopy (*Hamling et al., 2023*). All experiments used 5 dpf larvae from the *Tg(isl1:GFP);Tg(–6.7Tru.Hcrtr2:GAL4-VP16);Tg(UAS:GCaMP6s)* background. Briefly, larvae were mounted dorsal-up in 2% low-melt agarose in E3 onto a large beam diameter galvanometer system (ThorLabs GVS011). Tonic pitch-tilt stimuli were presented over a 65 s period in the following order: horizontal baseline (5 s at 0°), nose-down tilt (15 s at –19°), horizontal imaging (15 s at 0°), nose-up tilt (15 s at 19°), and horizontal imaging (15 s at 0°). Impulse stimuli contained a 4 ms eccentric rotation, a 2 ms hold, and a 4 ms restoration step to horizontal and were presented twice over a 65 s imaging window: horizontal baseline (20 s), impulse (10 ms), horizontal imaging (30 s), impulse (10 ms), horizontal imaging (15 s). Tonic and impulse stimuli were presented in alternating sets (impulse, then tonic) with a total of three stimulus set repeats.

Imaging was performed using a 20 x water immersion objective (Olympus XLUMPLFLN20xW 20 x/1.0), an infrared laser (SpectraPhysics MaiTai HP) at 920 nm using 6.1–18.8 mW of power at the sample, and ThorLabs LS 3.0 software. Experiments were conducted in the dark. High-resolution anatomy scans of nIII/nIV motor neurons were performed for each experiment to validate allele designations. Scans used a 147x147 µm imaging window, a 90 µm stack sampled every 1.5 µm, and a 5.2 microsecond pixel dwell time. Anatomy scans of the tangential nucleus were acquired using a 148x91 µm imaging window as a 40–50 µm stack sampled every 1 µm. For stimulus imaging, the tangential nucleus was sampled every 3–6 µm based on cell density. 6–10 planes were sampled for each hemisphere. Ventral planes were imaged at higher magnification (112x68 µm imaging window) than dorsal planes (148x91 µm window) to avoid photomultiplier tube saturation from in-frame GFP fluorescence; magnification was corrected for in later analyses. Laser power was adjusted for each sampled plane due to the light scattering properties of zebrafish tissue. As greater power was required for ventral planes, imaging was always performed from ventral to dorsal to minimize photobleaching effects. Stimulus imaging was performed at 3 frames/second (2.2 µs pixel dwell time) with a total time of approximately two hours per fish.

The number of neurons sampled per fish for each genotype are as follows (n=mean/standard deviation per fish; total cells/total fish): (1) Wildtype reference dataset: n=22±14 neurons/fish, N=255/10 total; (2) *phox2a*$^{+/+}$: 33±13 neurons/fish, N=76/5 total; (3) *phox2a*$^{+/-}$: 43±9 neurons/fish, N=109/6 total; (4) *phox2a*$^{-/-}$: 35±14 neurons/fish, N=297/16 total.

## Retrograde photolabeling of tangential nucleus neurons

Experiments were performed as described in *Goldblatt et al., 2023* based on *Pujala and Koyama, 2019* on 5 dpf larvae from the *Tg(isl1:GFP);Tg(–6.7Tru.Hcrtr2:GAL4-VP16);Tg(UAS-E1b:Kaede)* background. Briefly, experiments leveraged a photoconvertible protein, Kaede, which irreversibly converts from green to red with ultraviolet light. Larvae were raised in darkness to minimize background conversions. Larvae were mounted dorsal-up in 2% agarose under a confocal microscope. An imaging window was centered over the medial longitudinal fasciculus (MLF) and repeatedly scanned with a 405 nm laser for 30 s until fully converted (green to red). Off-target photoconversion was assessed (e.g. conversion of projections lateral to the MLF). Larvae were unmounted, left to recover in E3 for 4 hr in darkness, and then re-mounted in a lateral mount. An imaging window was centered around the tangential nucleus (see: Confocal Imaging). Retrogradely labeled soma were identified by their center-surround fluorescence appearance: red converted cytoplasm surrounding an unconverted green nucleus.

## Neuron harvesting, dissociation, and flow cytometry

Experiments were performed on 72–74 hpf larvae from the *Tg(isl1:GFP);Tg(–6.7Tru.Hcrtr2:GAL4-VP16);Tg(UAS-E1b:Kaede)* background. At 2 dpf, larvae were designated as null or sibling (wildtype/heterozygote) as described above. Three experimenters (D.G., K.R.H., and P.L) harvested neurons in parallel. Larvae were anesthetized in MESAB in Earle's Balanced Salt Solution with calcium, magnesium, and phenol red (EBSS, Thermo Fisher Scientific 24010043) and mounted dorsal-up in 2% agarose. Fluorescence in tangential nucleus neurons was visualized using a SugarCube LED Illuminator (Ushio America, Cypress CA) using 10 x eyepieces on a stereomicroscope (Leica Microsystems, Wetzlar, Germany). Neurons were harvested using a thin wall glass capillary tube (4 inch, OD 1.0 MM, World Precision Instruments) into EBSS in a non-stick Eppendorf tube and kept on ice until dissociation.

Neurons were dissociated in 20 units/mL of papain prepared in EBSS (Worthington Biochemical), 2000 units/mL of deoxyribonucleic prepared in EBSS (Worthington Biochemical), and 100 mg/mL of Type 1 A Collagenase (Sigma-Aldrich) prepared in Hanks Buffered Salt Solution without calcium/magnesium (HBSS, Thermo Fisher Scientific). Neurons were incubated for 45 minutes at 31.5 °C with a gentle vortex every 10–15 min, then passed through a 20 µm filter and centrifuged for 10 min at 300 x *g*. After removing supernatant, neurons were resuspended in L15 (Thermo Fisher Scientific) with 2% fetal bovine serum (Thermo Fisher Scientific). Cell health was evaluated using DAPI, applied at 0.5 µg/ml (Invitrogen) and incubated on ice for 30–45 min prior to flow cytometry.

Flow cytometry was performed using a Sony SH800z cell sorter (100 µm nozzle, 20 psi) to isolate single neurons (*Figure 5—figure supplement 1*). Three controls were run: (1) non-fluorescent wild-type neurons, (2) non-fluorescent neurons + DAPI, (3) fluorescent (green) neurons from *Tg(isl1:GFP);Tg(–6.7Tru.Hcrtr2:GAL4-VP16);Tg(UAS-E1b:Kaede)*+DAPI. On average, 2% of neurons were DAPI-positive and excluded. Neurons were evaluated for positive (green) fluorescence. Fluorescence was not evaluated to separate *Tg(UAS-E1b:Kaede)* neurons from those labeled by *Tg(isl1:GFP)*. Neurons were sorted into an Eppendorf tube containing 700 µl of lysis buffer (RNAqueous Micro Total RNA Isolation Kit, Thermo Fisher Scientific) for downstream bulk RNA sequencing.

## Bulk RNA sequencing

RNA isolation was performed using an RNAqueous Micro Total RNA Isolation Kit (Thermo Fisher Scientific). RNA concentration and quality (RIN > 8.0) was evaluated using an RNA 6000 Pico Kit and a 2100 Bioanalyzer system (Agilent Technologies, Santa Clara, California). RNA sequencing was performed by the NYU Genome Technology Center. Libraries were prepared using the low-input Clontech SMART-Seq HT with Nxt HT kit (Takara Bio USA) and sequenced using an Illumina NovaSeq 6000 with an S1 100 Cycle Flow Cell (v1.5).

## Quantification and statistical analysis

### Cell counting and spatial mapping of nIII/nIV motor neurons

Analysis was performed in Fiji/ImageJ (*Schindelin et al., 2012*) using the Cell Counter plugin. Anatomical stacks of nIII/nIV were subdivided in the dorsoventral axis as described in *Greaney et al., 2017* to facilitate localization. A point ROI was dropped over each neuron in the plane in which the soma

was brightest (center). The number of neurons in each dorsoventral plane and their coordinates were recorded. Neuron coordinates were standardized relative to a (0,0) point, defined as one corner of a standard-sized rectangular box centered over the extent of nIII/nIV in a maximum intensity projection. Differences in spatial location across genotypes was evaluated separately for each spatial axis using a two-tailed, two-sample Kolmogorov-Smirnov test. Probability distributions for figures were generated using the mean and standard deviation from bootstrapped data (n=100 iterations) to ensure results were robust to data from single larva.

### Analysis of calcium imaging experiments

Analysis methods are detailed in *Goldblatt et al., 2023* and summarized briefly here. Regions of Interest (ROIs) were drawn around tangential nucleus neurons for each stimulus plane sampled and adjusted for minor movement (1–2 µm) between trials. Raw fluorescence traces were extracted using Matlab R2020b (MathWorks, Natick, Massachusetts) and normalized by ROI size to account for variation in magnification. A neuron's response to tonic or impulse stimuli was defined as the change in fluorescence in the first second of restoration to horizontal following tilt delivery. Responses were normalized using a baseline period, defined as the mean fluorescence across the initial baseline window (5 s) preceding the nose-down tilt (nose-down response) or the last 3 s of the horizontal restoration following nose-down tilt (nose-up response). This was used to generate a ΔF/F value. A ΔF/F response was defined as significant if it was greater than two standard deviations above baseline. Directional selectivity was assigned by normalizing the difference in ΔF/F responses to each tilt by their sum. This generated a scale of values of ±1 (i.e. positive values represent nose-up selectivity; negative values, nose-down). Some neurons responded to both tilt directions with high similarity; we set a minimum threshold of abs(0.1) to distinguish neurons with a clear directional selectivity from untuned neurons.

### Spatial mapping of tangential nucleus neurons

Analysis methods are detailed in *Goldblatt et al., 2023* and summarized briefly here. All imaged neurons were manually registered to a reference framework using Adobe Illustrator (2021). Anatomy stacks from all experiments were aligned in the XY (rostrocaudal, mediolateral) axes using established anatomical landmarks (e.g. Mauthner cell body, medial longitudinal fasciculus, otic capsule). For Z-registration (dorsoventral axis), stacks were subdivided into eight sections using landmarks within and around the tangential nucleus (e.g. Mauthner cell body, neuropil). All registered images were verified by two independent observers (D.G. and S.H.). Neurons were localized to one dorsoventral section and a reference circle, representing a cell, was placed in Illustrator. Coordinates for each reference circle were recorded and standardized to an absolute (0,0) point (dorsomedial-most point of the tangential nucleus). Coordinates were imported into Matlab (R2020b) and used to generate a spatial map of imaged neurons.

### Statistical analysis of differences in tilt responses across *phox2a* genotypes

Statistical comparisons of tonic and impulse tilt responses are summarized in *Table 1*. Analyses used a one-way ANOVA with multiple comparisons. No significant differences (tonic tilt responses) or small differences (impulse responses) were observed across genotypes. Control data reported in Results and *Figures 2 and 3* is an aggregate from wildtype, *phox2a*^+/+^, and *phox2a*^+/-^larvae.

### Alignment, quality control, and differential expression analysis of bulk sequencing data

Initial alignment and analyses were performed by the Applied Bioinformatics Laboratories at the NYU School of Medicine (RRID:SCR_019178). Sequencing data was aligned to the GRCz11 zebrafish reference genome and two fluorescent markers (Kaede, GFP; NCBI). Eight datasets from four experimental repeats were aligned: four from *phox2a* mutants, and four from sibling controls. One experimental repeat had significantly higher variance in the first and second principal components, likely due to poor quality leading to extremely low transcript counts, and was excluded from downstream analyses. Number of cells/larvae sequenced and used in downstream analysis are as follows: Repeat 1, n=532/ n=904 cells from N=28/N=28 *phox2a* null/control larvae; Repeat 2, n=802/n=683 cells from N=27/

N=26 *phox2a* null/control larvae; Repeat 3, n=1000/n=1007 cells from N=41/N=40 *phox2a* null/control larvae; Repeat 4 (excluded): n=690/n=571 cells from N=33/N=33 *phox2a* null/control larvae Differential gene expression between conditions (*phox2a* mutants vs. sibling controls) was assessed using DESeq2 (*Love et al., 2014*). Differentially-expressed candidate genes met two criteria: $\log_2$ fold change >—2— and p adjusted < 0.05.

## Filtering of bulk sequencing data using a reference single-cell sequencing dataset

Analyses were performed in R. Detection of markers for motor neurons (*isl1, isl2a, isl2b*) (*Pfaff et al., 1996; Tokumoto et al., 1995*) and neurons caudal (*hoxd4a*) (*Prince et al., 1998; Moens and Prince, 2002*) and lateral (*barhl2*) (*Kinkhabwala et al., 2011*) to rhombomeres 4–6 supported that our dataset included other populations. We applied a filter to exclude erroneous gene expression from non-tangential nucleus populations.

Filtering was performed using an existing single-cell atlas of neurons labeled in *Tg(–6.7Tru.Hcrtr2:GAL4-VP16);(Tg(UAS-E1b:Kaede))*, generated with 10 x Genomics. The reference atlas was generated from four experimental samples using the harvest, dissociation, and flow cytometry method described above. The sequenced atlas contained 1,468 neurons (*Figure 5—figure supplement 2A, B*). Data was analyzed using Seurat v4.0 (*Hao et al., 2021*). Cluster annotation was performed using a combination of fluorescent in situ hybridization as described above (*Figure 5—figure supplement 2C–E* and other data not shown) and published molecular data of the zebrafish hindbrain (*Moens and Prince, 2002*). n=159 neurons (11%) were validated as excitatory projection neurons from the tangential nucleus.

Genes in the bulk dataset were only included in downstream analyses if they were expressed above threshold percent of reference projection neurons: 1%, 3%, 5%, 10%, 30%, or 50%. The most stringent filter (50%) was set using the transcription factor *evx2*, which is reported to be expressed in all tangential nucleus neurons (*Sugioka et al., 2023*) and was detected in 36% of reference projection neurons. Qualitatively, we found that gene detection with fluorescent in situ hybridization scaled with reference filter stringency (*Figure 5—figure supplement 3*). Analyses were performed separately for each threshold. The total number of genes included for downstream analyses for each threshold are as follows: 28,807 (no threshold), 11,278 (1% of reference neurons), 8,346 (3%), 6,701 (5%), 4,185 (10%), 1,105 (30%), 384 (50%). We used the following significance thresholds for differential gene expression in filtered datasets: adjusted p value < 0.05 and abs(log2FoldChange) > 2. The number of differentially expressed genes for each threshold was as follows: 100 (no threshold), 23 (1% of reference neurons), 6 (3%), 2 (5%), 0 (10%).

Projection neurons in the tangential nucleus lie in close proximity to the medial vestibular nucleus (3–5 µm) to the medial edge of the tangential nucleus and 10–20 µm dorsal to the rostral/dorsal edge of the tangential nucleus. Some MVN neurons express *phox2a* (*Figure 5—figure supplement 5*). Our reference single-cell atlas isolated a small cluster of *phox2a*-expressing neurons in r5-6 that likely originates from this population. To control for the possibility that some differentially expressed genes are localized to the MVN, and not projection neurons, we also evaluated differential gene expression in the *phox2a*-expressing subset of MVN neurons (n=45 neurons, 3% of reference dataset). Data is shown in *Figure 5—figure supplement 5*.

## Generation of representative images for fluorescent in situ hybridization

Images were generated using Fiji/ImageJ (*Schindelin et al., 2012*). An anatomical template of the tangential nucleus was generated based on *Goldblatt et al., 2023*. Briefly, for sagittal view images, a 30 µm stack was centered over the tangential nucleus. For each plane, a region of interest (ROI) was drawn over all cells within the bounds of the tangential nucleus. Transcript expression outside the ROI was masked. Maximum intensity projections were generated. Minimal or no alterations to brightness/contrast were made for probe expression given the correlation between fluorescence intensity and detected transcript (*Choi et al., 2018*).

## Additional statistics

Bias and variability in probability distributions were estimated by bootstrapping, or resampling the raw distributions with replacement (*Efron and Tibshirani, 1986*). Data shown is the mean and standard

deviation of 100 bootstrapped distributions. Topography data was evaluated using two-tailed, two-way Kolmogorov-Smirnov tests. Functional responses to tilts (i.e. calcium response strength, direction-ality index) were evaluated using two-tailed Wilcoxon rank sum tests. Differences in responses across genotypes were analyzed using one-way ANOVA tests.

## Acknowledgements

Research was supported by the National Institute on Deafness and Communication Disorders of the National Institutes of Health under award numbers R01DC017489, F31DC020910, and F31DC019554, the National Institute of Neurological Disorders and Stroke under award numbers F99NS129179, T32NS086750, and the National Cancer Institute P30CA016087. The authors would like to thank Hannah Gelnaw for assistance with fish care, and Jeremy Dasen, Claude Desplan, Katherine Nagel, Dan Sanes, along with the members of the Schoppik and Nagel labs for their valuable feedback and discussions. Finally, the authors gratefully acknowledge the late Hans Straka for his generous insights and encouragement throughout.

## Additional information

### Funding

| Funder | Grant reference number | Author |
|---|---|---|
| National Institute on Deafness and Other Communication Disorders | R01DC017489 | David Schoppik |
| National Institute on Deafness and Other Communication Disorders | F31DC020910 | Paige Leary |
| National Institute on Deafness and Other Communication Disorders | F31DC019554 | Kyla Rose Hamling |
| National Institute of Neurological Disorders and Stroke | F99NS129179 | Dena Goldblatt |
| National Institute of Neurological Disorders and Stroke | T32NS086750 | Dena Goldblatt |

The funders had no role in study design, data collection and interpretation, or the decision to submit the work for publication.

### Author contributions

Dena Goldblatt, Conceptualization, Funding acquisition, Investigation, Visualization, Methodology, Writing - original draft; Basak Rosti, Kyla Rose Hamling, Paige Leary, Investigation, Methodology; Harsh Panchal, Marlyn Li, Hannah Gelnaw, Stephanie Huang, Cheryl Quainoo, Investigation; David Schoppik, Conceptualization, Supervision, Funding acquisition, Methodology, Writing – review and editing

### Author ORCIDs

Dena Goldblatt ⬥ https://orcid.org/0000-0003-0333-2433
Paige Leary ⬥ https://orcid.org/0000-0002-0888-466X
David Schoppik ⬥ https://orcid.org/0000-0001-7969-9632

### Ethics

All protocols and procedures involving zebrafish were approved by the New York University Langone School of Medicine Institutional Animal Care & Use Committee (IACUC; approval number IA16-00561).

Reviewer #1 (Public review): https://doi.org/10.7554/eLife.96893.3.sa1

Reviewer #2 (Public review): https://doi.org/10.7554/eLife.96893.3.sa2
Reviewer #3 (Public review): https://doi.org/10.7554/eLife.96893.3.sa3
Author response https://doi.org/10.7554/eLife.96893.3.sa4

## Additional files

### Supplementary files
• MDAR checklist

### Data availability
All data and code are deposited at the Open Science Framework and are publicly available at https://doi.org/10.17605/OSF.IO/93V6E. Sequencing data has been deposited in GEO under accession codes GSE254346 and GSE254345.

The following datasets were generated:

| Author(s) | Year | Dataset title | Dataset URL | Database and Identifier |
|---|---|---|---|---|
| Goldblatt D, Rosti B, Hamling KR, Leary P, Panchal H, Li M, Gelnaw H, Huang S, Quainoo C, Schoppik D | 2024 | Effect of phox2a knockout on the molecular profiles of hindbrain vestibular neurons in the larval zebrafish (bulk RNA-Seq) | https://www.ncbi.nlm.nih.gov/geo/query/acc.cgi?acc=GSE254345 | NCBI Gene Expression Omnibus, GSE254345 |
| Goldblatt D, Hamling KR, Leary P, Li M, Panchal H, Gelnaw H, Huang S, Schoppik D | 2024 | Molecular characterization of hindbrain vestibular neurons in the larval zebrafish (scRNA-Seq) | https://www.ncbi.nlm.nih.gov/geo/query/acc.cgi?acc=GSE254346 | NCBI Gene Expression Omnibus, GSE254346 |

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
