## [Editor Report · eLife Assessment]

This **important** study asks whether motor neurons within the vestibulo-ocular circuit of zebrafish are required to determine the identity, connectivity, and function of upstream premotor neurons. They provide **compelling** and comprehensive genetic, anatomical and behavioral evidence that the answer is, "No!". This work will be of general interest to developmental neurobiologists and will motivate future studies of whether motor neurons are dispensable for assembly of other sensorimotor neural circuits.

---

## [Referee Report · Reviewer #1 (Public review)]

Summary:

This study has as its goal to determine how the structure and function of the circuit that stabilizes gaze in the larval zebrafish depends on the presence of the output cells, the motor neurons. A major model of neural circuit development posits that the wiring of neurons is instructed by their postsynaptic cells, transmitting signals retrogradely on which cells to contact and, by extension, where to project their axons. Goldblatt et al. remove the motor neurons from the circuit by generating null mutants for the phox2a gene. The study then shows that, in this mutant that lacks the isl1-labelled extraocular motor neurons, the central projection neurons have (1) largely normal responses to vestibular input; (2) normal gross morphology; (3) minimally changed transcriptional profiles. From this, the authors conclude that the wiring of the circuit is not instructed by the output neurons, refuting the major model.

Strengths:

I found the manuscript to be exceptionally well-written and presented, with clear and concise writing and effective figures that highlight key concepts. The topic of neural circuit wiring is central to neuroscience, and the paper's findings will interest researchers across the field, and especially those focused on motor systems.

The experiments conducted are clever and of a very high standard, and I liked the systematic progression of methods to assess the different potential effects of removing phox2a on circuit structure and function. Analyses (including statistics) are comprehensive and appropriate and show the authors are meticulous and balanced in most of the conclusions that they draw. Overall, the findings are interesting and should leave little doubt about the paper's main conclusions.

Weaknesses:

All conclusions are supported by the data, and the characterisation of the effects of the main manipulation in the study, removing phox2a to take out the extra-ocular motor neurons, is extensive. I cannot see weaknesses that affect the conclusions in this manuscript.

The study raises interesting questions that could be addressed in future work, which would further explain how the projection neurons develop. While the cells that would have been extraocular motor neurons are still there in phox2a mutants, they can no longer be called motor neurons as they lack expression of vachta and isl1. It would therefore be interesting to see what an alternative manipulation, e.g., the physical removal of the motor neurons using laser ablation, would have. Furthermore, the motor neurons are dispensable for the projection neurons' wiring, but the projection neurons innervate several other cell types that could affect their development. A future project could determine the precise contribution of each postsynaptic population on the projection neurons' development.

---

## [Referee Report · Reviewer #2 (Public review)]

Summary:

This study was designed to test the hypothesis that motor neurons play a causal role in circuit assembly of the vestibulo-ocular reflex circuit, which is based on the retrograde model proposed by Hans Straka. This circuit consists of peripheral sensory neurons, central projection neurons, and motor neurons. The authors hypothesize that loss of extraocular motor neurons, through CRISPR/Cas9 mutagenesis of the phox2a gene, will disrupt sensory selectivity in presynaptic projection neurons if the retrograde model is correct.

Account of the major strengths and weaknesses of the methods and results:

The work presented is impressive in both breadth and depth, including the experimental paradigms. Overall, the main results were that the loss of function paradigm to eliminate extraocular motor neurons did not (1) alter the normal functional connections between peripheral sensory neurons and central projection neurons, (2) affect the position of central projection neurons in the sensorimotor circuit, or (3) significantly alter the transcriptional profiles of central projection neurons. Together, these results strongly indicate that retrograde signals from motor neurons are not required for the development of the sensorimotor architecture of the vestibulo-ocular circuit.

Appraisal of whether the authors achieved their aims, and whether the results support their conclusions:

The results of this study showed that extraocular motor neurons were not required for central projection neuron specification in the vestibulo-ocular circuit, which countered the prevailing retrograde hypothesis proposed for circuit assembly. A concern is that the results presented may be limited to this specific circuit and may not be generalizable to other circuit assemblies, even to other sensorimotor circuits.

Discussion of the likely impact of the work on the field, and the utility of the methods and data to the community:

As mentioned above, this study sheds valuable new insights into the developmental organization of the vestibulo-ocular circuit. However, different circuits likely utilize various mechanisms, extrinsic or intrinsic (or both), to establish proper functional connectivity. So, the results shown here, although they begin to explain the developmental organization of the vestibulo-ocular circuit, whether generalizable to other circuits is debatable. At a minimum, this study provides a starting point for the examination of the patterning of connections in this and other sensorimotor circuits.

---

## [Referee Report · Reviewer #3 (Public review)]

In this manuscript by Goldblatt et al. the authors study the development of a well-known sensorimotor system, the vestibulo-ocular reflex circuit, using *Danio rerio* as a model. The authors address whether motor neurons within this circuit are required to determine the identity, upstream connectivity and function of their presynaptic partners, central projection neurons. They approach this by generating a CRISPR-mediated knockout line for the transcription factor phox2a, which specifies the fate of extraocular muscle motor neurons. After showing that phox2a knockout ablates these motor neurons, the authors show that functionally, morphologically, and transcriptionally, projection neurons develop relatively normally.

Overall, the authors present a convincing argument for the dispensability of motor neurons in the wiring of this circuit, although their claims about the generalizability of their findings to other sensorimotor circuits should be tempered. The study is comprehensive and employs multiple methods to examine the function, connectivity and identity of projection neurons.

Comments on the revised version:

The authors have addressed all my previous concerns.

---

## [Author Response]

The following is the authors’ response to the original reviews.

**Reviewer 1:**
SummaryThis study has as its goal to determine how the structure and function of the circuit that stabilizes gaze in the larval zebrafish depends on the presence of the output cells, the motor neurons. A major model of neural circuit development posits that the wiring of neurons is instructed by their postsynaptic cells, transmitting signals retrogradely on which cells to contact and, by extension, where to project their axons. Goldblatt et al. remove the motor neurons from the circuit by generating null mutants for the phox2a gene. The study then shows that, in this mutant that lacks the isl1-labelled extraocular motor neurons, the central projection neurons have (1) largely normal responses to vestibular input; (2) normal gross morphology; (3) minimally changed transcriptional profiles. From this, the authors conclude that the wiring of the circuit is not instructed by the output neurons, refuting the major model.StrengthsI found the manuscript to be exceptionally well-written and presented, with clear and concise writing and effective figures that highlight key concepts. The topic of neural circuit wiring is central to neuroscience, and the paper's findings will interest researchers across the field, and especially those focused on motor systems.The experiments conducted are clever and of a very high standard, and I liked the systematic progression of methods to assess the different potential effects of removing phox2a on circuit structure and function. Analyses (including statistics) are comprehensive and appropriate and show the authors are meticulous and balanced in most of the conclusions that they draw. Overall, the findings are interesting, and with a few tweaks, should leave little doubt about the paper's main conclusions.

We are grateful for the Reviewer’s enthusiasm for our manuscript and recognition of the advance to the vestibular and motor systems fields. We particularly appreciate their suggestions for experiments to improve the characterization of our *phox2a* mutant line. We hope the Reviewer finds the results of the added experiment adequately address the points they raise.

Weaknesses/Recommendations(1) The main point is the incomplete characterisation of the effects of removing phox2a on the extra-ocular motor neurons. Are these cells no longer there, or are they there but no longer labelled by isl1:GFP? If they are indeed removed, might they have developed early on, and subsequently lost? These questions matter as the central focus of the manuscript is whether the presence of these cells influences the connectivity and function of their presynaptic projection neurons. Therefore, for the main conclusions to be fully supported by the data, the authors would need to test whether (1) the motor neurons that otherwise would have been labelled by the isl1:GFP line are physically no longer there; (2) that this removal (if, indeed, it is that) is developmental. If these experiments are not feasible, then the text should be adjusted to take this into account.Show (e.g., with DAPI or some other staining) whether there are still cells where you would have expected to see nIII/nIV extraocular motor neurons. If this is done in a developmental timeline both main "concerns" are addressed in one go. If this doesn't work for some reason, then I'd suggest adjusting the discussion section to note this caveat. I realise it is commonplace in zebrafish and rodent papers to equate the two, but it should also be considered that the isl1:GFP does not report which cells are isl1+ 100% faithfully.

We thank the Reviewer for their suggestion. We’ve included the results of this experiment in (new) Supplemental Figure 1 and have updated the Results accordingly (text lines 69-72).

Briefly: We performed fluorescent *in situ* hybridization for *vachta*, a marker for cholinergic motor neurons, when nIII/nIV differentiation is complete at 2 dpf and prior to synaptogenesis with both their pre- and postsynaptic partners. We included a DAPI stain. We find that while *phox2a* does not physically remove neurons from the region that contains nIII/nIV motor neurons, neurons in this region no longer express *vachta*. The presence of neurons at an early stage (2dpf) that have lost expression of both a transcription factor (*isl1*) and motor neuron marker (*vachta*) supports our contention that, while cells are there, they should not be considered motor neurons.

While the reviewer did not suggest it directly, we note that there is a more laborious way to determine “what happens to cells that would have been *phox2a+* but no longer express *phox2a?”* Specifically, one could target a reporter transgene to the endogenous *phox2a* locus on the *phox2a* mutant background. Regrettably, generating such a knock-in reporter is difficult and success is far from assured.

Previously (Greaney et. al. 2017, 10.1002/cne.24042), we compared expression patterns in nIV to those observed after retro-orbital dye fills. We never saw neurons labeled by dye that were not also GFP+. However, it was not possible to perform a similar analysis for nIII, so we acknowledge the limits of the isl1:GFP reporter.

(2) A further point to address is the context of the manipulation. If the phox2a removal does indeed take out the extra-ocular motor neurons, what percentage of postsynaptic neurons to the projection neurons are still present?In other words, how does the postsynaptic nMLF output relate to the motor neurons? If, for instance, the nMLF (which, as the authors state, are likely still innervated by the projection neurons) are the main output of the projections neurons, then this would affect the interpretation of the results.Is there quantitative information on the projection neuron outputs to address the second point (i.e., how much of the projection neurons' output is the extra-ocular motor neurons)? If not, it should be discussed how this could affect the conclusions.

Qualitatively, projection neurons form more robust arbors to the nMLF than to their nIII/nIV partners (see: Schoppik et al. 2017, DOI: 10.1523/JNEUROSCI.1711-17.2017). We expect this is proportional to the size of each downstream target.

The Reviewer makes an interesting point here. These projection neurons innervate several downstream nuclei that could potentially influence their development; we’ve considered this in the Discussion based on existing literature and in the context of our own findings. A precise dissection of each target population’s contribution would be interesting and important for larger questions about neural circuits for balance (see Sugioka et. al. 2023 10.1038/s41467-023-36682-y). However, we feel this analysis is outside our study’s scope, given that our aim here was to evaluate a standing hypothesis restricted to the contribution of nIII/nIV motor neurons.

Less important, but still useful:- Figure 4C/D: I found these panels difficult to interpret. Perhaps split them up so each panel does a little less heavy lifting? Do the main panels in C show all axons? Where are the "two remaining nIII/nIV neurons" in D?

We’ve split the panels in 4C as suggested and adjusted the caption text in 4D to clarify the “remaining neurons” were simply not eliminated following *phox2a* knockout. We presume they are instead *phox2b*+. 4C shows all axons labeled by our transgenic line that follow the medial longitudinal fasciculus.

Extremely minor:- line 28: "tantamount"  "paramount"?- some figure legends say DeltaFF, instead of DeltaF/F- line 192: "the any"

These have been corrected; we thank the reviewer for their attention to detail.

**Reviewer 2:**
SummaryThis study was designed to test the hypothesis that motor neurons play a causal role in circuit assembly of the vestibulo-ocular reflex circuit, which is based on the retrograde model proposed by Hans Straka. This circuit consists of peripheral sensory neurons, central projection neurons, and motor neurons. The authors hypothesize that loss of extraocular motor neurons, through CRISPR/Cas9 mutagenesis of the phox2a gene, will disrupt sensory selectivity in presynaptic projection neurons if the retrograde model is correct.StrengthsThe work presented is impressive in both breadth and depth, including the experimental paradigms. Overall, the main results were that the loss of function paradigm to eliminate extraocular motor neurons did not (1) alter the normal functional connections between peripheral sensory neurons and central projection neurons, (2) affect the position of central projection neurons in the sensorimotor circuit, or (3) significantly alter the transcriptional profiles of central projection neurons. Together, these results strongly indicate that retrograde signals from motor neurons are not required for the development of the sensorimotor architecture of the vestibulo-ocular circuit.

We are grateful for the excellent summary of our manuscript and support for our aim, which was indeed to evaluate Hans Straka’s model for the development of the vestibulo-ocular reflex circuit.

Appraisal of whether the authors achieved their aims, and whether the results support their conclusions The results of this study showed that extraocular motor neurons were not required for central projection neuron specification in the vestibulo-ocular circuit, which countered the prevailing retrograde hypothesis proposed for circuit assembly. A concern is that the results presented may be limited to this specific circuit and may not be generalizable to other circuit assemblies, even to other sensorimotor circuits.ImpactAs mentioned above, this study sheds valuable new insights into the developmental organization of the vestibulo-ocular circuit. However, different circuits likely utilize various mechanisms, extrinsic or intrinsic (or both), to establish proper functional connectivity. So, the results shown here, although begin to explain the developmental organization of the vestibulo-ocular circuit, are not likely to be generalizable to other circuits; though this remains to be seen. At a minimum, this study provides a starting point for the examination of patterning of connections in this and other sensorimotor circuits.Weaknesses/RecommendationsA concern is that the results presented may be limited to this specific circuit and may not be generalizable to other circuit assemblies, even to other sensorimotor circuits. However, different circuits likely utilize various mechanisms, extrinsic or intrinsic (or both), to establish proper functional connectivity. So, the results shown here, although begin to explain the developmental organization of the vestibulo-ocular circuit, are not likely to be generalizable to other circuits; though this remains to be seen.

We agree with the Reviewer that — of course — a diverse array of developmental mechanisms shape sensorimotor circuit architecture. However, prior findings in the spinal cord (Wang & Scott 2000, Sürmeli et al. 2011, Bikoff et al. 2016, Sweeney et al. 2018, Shin et al. 2020) support our primary conclusion that motor neurons are dispensable for specification of premotor partners. The Recommendation ends with “though this remains to be seen.” We infer that the Reviewer does not have a counterexample at hand for a circuit where motor neurons determine the fate of their partners. Therefore, the preponderance of evidence argues that our work is likely to generalize to other circuits. However, we acknowledge the limitations of our work and we have tempered any claims to generality in the text.

Lines 156-57: The authors should consider and discuss explicitly the potential of compensatory mechanisms in the CRISPR/Cas9 mutants that may permit synaptogenesis of the projection neurons even though MNs partners are absent.

We agree with the Reviewer that careful consideration of compensation is merited when using mutants. There are two synapses that the comment might refer to: those between projection neurons and motor neurons, and those between sensory afferents and projection neurons. Projection neurons fail to form any synapses at the region that would contain their motor neuron (nIII/nIV) partners (see Fig. 4C), so there is no question of compensation there. Figure 1B shows that there is no *phox2a* expression in sensory or central projection neurons. Consequentially, even if there were a gene that perfectly compensated for the loss of *phox2a* it wouldn’t be active in sensory or central projection neurons. We therefore do not believe that compensatory expression of other genes plays any role here.

Line 162: Is this an accurate global statement or should it be restricted to the evidence provided in this report?

We’ve clarified this line, which summarizes findings described in previous results sections of this report.

**Reviewer 3:**
SummaryIn this manuscript by Goldblatt et al. the authors study the development of a well-known sensorimotor system, the vestibulo-ocular reflex circuit, using *Danio rerio* as a model. The authors address whether motor neurons within this circuit are required to determine the identity, upstream connectivity and function of their presynaptic partners, central projection neurons. They approach this by generating a CRISPR-mediated knockout line for the transcription factor phox2a, which specifies the fate of extraocular muscle motor neurons. After showing that phox2a knockout ablates these motor neurons, the authors show that functionally, morphologically, and transcriptionally, projection neurons develop relatively normally.Overall, the authors present a convincing argument for the dispensability of motor neurons in the wiring of this circuit, although their claims about the generalizability of their findings to other sensorimotor circuits should be tempered. The study is comprehensive and employs multiple methods to examine the function, connectivity and identity of projection neurons.

We appreciate the Reviewer’s support for our manuscript and have implemented their thoughtful suggestions on how to improve the clarity and presentation of our conclusions. We acknowledge the shared consideration with Reviewer 2 as to the generalizability of our findings, and have tempered the language in our revision.

In the introduction the authors set up the controversy on whether or not motor neurons play an instructive role in determining "pre-motor fate". This statement is somewhat generic and a bit misleading as it is generally accepted that many aspects of interneuron identity are motor neuron-independent. The authors might want to expand on these studies and better define what they mean by "fate", as it is not clear whether the studies they are citing in support of this hypothesis actually make that claim.

We appreciate the Reviewer’s attention to this important consideration. We agree that there are numerous, and often ambiguous ways to define cell fate. We’ve modified our manuscript to read “…for and against an instructive role in establishing connectivity” (line 19) to reflect that connectivity is the most pertinent readout of cell fate in (most) studies cited there, as well as in our model system (lines 25-26: “Subtype fate, anatomical connectivity, and function are inextricably linked: directionally-tuned sensory neurons innervate nose-up/nosedown subtypes of projection neurons, which in turn innervate specific motor neurons…”). We’ve expanded on the prior studies mentioned above in relevant sections of the Results and Discussion.

Although it appears unchanged from their images, the authors do not explicitly quantitate the number of total projection neurons in phox2a knockouts.

We have added this quantification (text lines 95-96); the number of projection neurons per hemisphere is unchanged in control and mutant larvae.

For figures 2C and 3C, please report the proportion of neurons in each animal, either showing individual data points here or in a separate supplementary figure; and please perform and report the results of an appropriate statistical test.

Generally, we agree that per-animal sampling can provide important metrics. We’ve added a line in the appropriate Methods section with the mean/standard deviation number of neurons sampled per animal for each genotype (lines 408-410). However, our extensive prior work using this transgenic line (Goldblatt et al. 2023, DOI: 10.1016/j.cub.2023.02.048) argued that a per-animal breakdown can be misleading. Due to occasional technical aberrations, variation in transgenic line expression, and limitations of our registration method, we cannot sample 100% of the projection nucleus (~50 neurons/hemisphere) in each animal. Likewise, the topography of the nucleus in WT animals, both for up/down subtypes (Fig. 2) and impulse responsive/unresponsive neurons (Fig. 3), means that subtypes may not be proportionally sampled on a peranimal basis. While such problems would likely resolve if we took data from ~50-75 animals for each condition, at a throughput of ~2 animals/day and 1-2 experimental days / week on shared instrumentation the throughput simply isn’t there. We therefore believe the data is best represented as an aggregate.

In the topographical mapping of calcium responses (figures 2D, E and 3D), the authors say they see no differences but this is hard to appreciate based on the 3D plotting of the data. Quantitating the strength of the responses across the 3-axes shown individually and including statistical analyses would help make this point, especially since the plots look somewhat qualitatively different.

We have added a supplemental table (new Table 2) with statistical comparisons of projection neuron topography (both to tonic and impulse stimuli) across genotypes for additional clarification. Quantitatively, we find that differences in projection neuron position (max observed: approx. 5 microns) are within the limits of our expected error in registering neurons across larvae to a standardized framework, given the small size of the nucleus (approx. 40 microns in each spatial axis) and each individual neuron (approx. 5 microns in diameter).

The transcriptional analysis is very interesting, however, it is not clear why it was performed at 72 hpf, while functional experiments were performed at 5 days. Is it possible that early aspects of projection neuron identity are preserved, while motor neuron-dependent changes occur later? The authors should better justify and discuss their choice of timepoint.

As suggested, we have updated the manuscript to justify the choice of timepoint (text lines 176-177). We agree with the Reviewer that choosing the “right” timepoint for transcriptional analysis is key. The comment underscores the challenges in balancing the amount of time past neurogenesis (24-54 hpf) when potential fate markers could change, with the timecourse of synaptogenesis (2-4 dpf) and functional maturation (5 dpf). We hypothesized that selecting an intermediate timepoint (72 hpf, during peak synaptogenesis), would enable the highest resolution of both fate markers expressed at the end of neurogenesis (54 hpf) and wiring specificity molecules. We point the Reviewer to recent studies in comparable systems that proposed subtype diversity is most resolvable during synaptogenesis as further justification (see: Ozel et al. 2022, DOI: 10.1038/s41586-020-2879-3 and Li et al. 2017, DOI: 10.1016/j.cell.2017.10.019). However, we acknowledge that the ideal experiment would have been a transcriptional timecourse that would have directly addressed the question.

The inclusion of heterozygotes as controls is problematic, given that the authors show there are notable differences between phox2a+/+ and phox2a+/- animals; pooling these two genotypes could potentially flatten differences between controls and phox2a-/-.

We agree that this is an important limitation on our interpretations and have noted this more explicitly in the appropriate Results section (line 204).

Projection neurons appear to be topographically organized and this organization is maintained in the absence of motor neurons. Are there specific genes that delineate ventral and dorsal projection neurons? If so, the authors should look at those as candidate genes as they might be selectively involved in connectivity. Showing that generic synaptic markers (Figure 4E) are maintained in the entire population is not convincing evidence that these neurons would choose the correct synaptic partners.

We agree with the Reviewer that Figure 4E is limited and that the most convincing molecular probe would be against a subtype-specific marker gene, ideally the one(s) that establish subtype-specific connectivity. To date, few such markers have been identified in any system, and, to the best of our knowledge, no reported markers differentiate dorsal (nose-up) from ventral (nose-down) projection neurons. We are currently evaluating candidates, but will not include that data here until the relevant genes are established as veridical subtype markers with defined roles in subtype fate specification and connectivity.